# STAT5B leukemic mutations, altering SH2 tyrosine 665, have opposing impacts on immune gene programs

Hye Kyung Lee[1], Jichun Chen[2], Rachael L Philips[3], Sung-Gwon Lee[1], Xingmin Feng[2], Zhijie Wu[2], Chengyu Liu[4], Aaron B Schultz[5,6], Molly Dalzell[5,6], Manja Meggendorfer[7], Claudia Haferlach[7], Foster Birnbaum[8,9], Joel A Sexton[8], Amy E Keating[8,10,11], John J O'Shea[3], Neal S Young[2], Alejandro V Villarino[5,6], Priscilla A Furth[1], Lothar Hennighausen[1]

**STAT5B is a vital transcription factor for lymphocytes. Here, the function of two STAT5B mutations from human T-cell leukemias: one substituting tyrosine 665 with phenylalanine (STAT5B^Y665F) and the other with histidine (STAT5B^Y665H), was interrogated. In silico modeling predicted divergent energetic effects on homo-dimerization with a range of pathogenicity. In primary T cells in vitro, STAT5B^Y665F showed gain-of-function, whereas STAT5B^Y665H demonstrated loss-of-function. Introducing the mutation into the mouse genome illustrated that the gain-of-function _Stat5b_^Y665F mutation resulted in accumulation of CD8^+ effector and memory and CD4^+ regulatory T cells, altering CD8^+/CD4^+ ratios. In contrast, STAT5B^Y665H "knock-in" mice showed diminished CD8^+ effector and memory and CD4^+ regulatory T cells. In contrast to WT STAT5B, the STAT5B^Y665F variant displayed greater STAT5 phosphorylation, DNA binding, and transcriptional activity after cytokine activation, whereas the STAT5B^Y665H variant resembled a null. The work exemplifies how joining in silico and in vivo studies of single nucleotides deepens our understanding of disease-associated variants, revealing structural determinants of altered function, defining mechanistic roles, and, specifically here, identifying a gain-of-function variant that does not directly induce hematopoietic malignancy.**

## Introduction

Transcription factors belonging to the signal transducers and activators of transcription (STAT) family are activated by upstream cytokines and growth factors and, in turn, regulate both universal and lineage-specific genetic programs (Philips et al, 2022). STAT5A (Wakao et al, 1994; Liu et al, 1995) and STAT5B (Azam et al, 1995; Liu et al, 1995; Mui et al, 1995), ancestral members of the STAT family, play defining roles within the hematopoietic system (Yao et al, 2006), the mammary gland (Liu et al, 1997; Miyoshi et al, 2001; Cui et al, 2004), body growth (Davey et al, 1999; Holloway et al, 2007; Hwa, 2021), and liver metabolism (Holloway et al, 2007). As "signal-dependent" transcription factors (TFs), they instruct common and lineage-restricted enhancers and super-enhancers (Yamaji et al, 2013; Shin et al, 2016; Villarino et al, 2016, 2017; Li et al, 2017; Lee et al, 2023), thereby controlling a wide range of genetic programs, in tissues as diverse as the mammary gland, liver, and the immune system.

Although genetic programs regulated by native STAT5A and STAT5B are well investigated, the impact of missense mutations on these TFs has not been studied in detail. Based on current literature and genomic databases, at least one third of the amino acids of STAT5 can acquire germline and somatic missense mutations in humans. Inactivating STAT5B germline mutations (Rajala et al, 2013; Hwa, 2021; Bhattacharya et al, 2022) have been associated with growth hormone insensitivity (Laron syndrome) and immune pathology, whereas activating somatic mutations (Rajala et al, 2013; Kiel et al, 2014; Kontro et al, 2014; Bhattacharya et al, 2022) have been identified in patients with T-cell large granular lymphoblastic leukemia (T-LGLL) (Rajala et al, 2013; Kiel et al, 2014; Kontro et al, 2014; Bhattacharya et al, 2022) and T-cell prolymphocytic leukemia (T-PLL) (Kiel et al, 2014). T-LGLL is a rare disorder marked by the clonal expansion of cytotoxic T cells within the peripheral blood and bone marrow (Matutes, 2018). Among the two subtypes of T-LGLL, the more frequent CD8 T-LGLL is characterized by STAT3 mutations, whereas the CD4 T-LGLL has been linked to mutations in the STAT5B, which are clustered in the SH2 and C-terminal

[1]Laboratory of Genetics and Physiology, National Institute of Diabetes and Digestive and Kidney Diseases, US National Institutes of Health, Bethesda, MD, USA [2]Hematology Branch, National Heart, Lung, and Blood Institute, National Institutes of Health, Bethesda, MD, USA [3]Molecular Immunology and Inflammation Branch, National Institute of Arthritis and Musculoskeletal and Skin Diseases, National Institutes of Health, Bethesda, MD, USA [4]Transgenic Core, National Heart, Lung, and Blood Institute, US National Institutes of Health, Bethesda, MD, USA [5]Department of Microbiology and Immunology, Miller School of Medicine, University of Miami, Miami, FL, USA [6]Sylvester Comprehensive Cancer Center, University of Miami, Miami, FL, USA [7]Munich Leukemia Laboratory (MLL) Max-Lebsche-Platz 31, München, Germany [8]Department of Biology, Massachusetts Institute of Technology, Cambridge, MA, USA [9]Computational and Systems Biology, Massachusetts Institute of Technology, Cambridge, MA, USA [10]Department of Biological Engineering, Massachusetts Institute of Technology, Cambridge, MA, USA [11]Koch Institute for Integrative Cancer Research, Massachusetts Institute of Technology, Cambridge, MA, USA

Correspondence: hyekyung.lee@nih.gov; priscilla.furth@nih.gov; lotharh@nih.gov

transactivation domain. Accordingly, N642H and Y665F are the most frequent STAT5B mutations, and both lead to increased STAT5 activity (Andersson et al, 2016; Bhattacharya et al, 2022).

The SH2 domain is central to cytokine-induced, JAK-dependent STAT5B tyrosine phosphorylation, activation, dimerization, nuclear translocation, and establishment of functional transcription enhancers controlling genetic programs. A total of six somatic missense mutations and four germline mutations within the SH2 domain of STAT5B have been reported in the literature (Hwa, 2021; Bhattacharya et al, 2022). Although cell culture experiments have established their capacity to function as gain-of-function (GOF) or loss-of-function (LOF), their complex physiological and genomic impacts have not been investigated. Of particular interest are two single nucleotide variants (SNV) identified in patients with T-LGLL and T-PLL (Kontro et al, 2014). Both are found within the SH2 domain and change tyrosine 665 (Y665) to either phenylalanine (Y665F) or histidine (Y665H); however, it should be noted that although the STAT5B[Y665F] mutation is reproducibly found, only one case of the STAT5B[Y665H] mutation was reported (in a T-PLL case) (Rajala et al, 2013; Kiel et al, 2014). Both mutations have been described as GOF in vitro (Pham et al, 2018), suggesting that they activate similar and possibly overlapping genetic programs. To determine whether they are driver mutations for T-cell leukemias, we integrated STAT5B[Y665F] and STAT5B[Y665H] into the mouse genome and examined their impact on lymphocyte development and function. Using this approach, we show that only the STAT5B[Y665F] mutation displays GOF characteristics in vivo, whereas STAT5B[Y665H] is a LOF variant. Significantly neither of them directly induced malignant transformation. These data expand our understanding of the structural and molecular impact of these two specific single nucleotide STAT5B variants in lymphocyte homeostasis and function, identifying specific genes and pathways involved in their pathogenesis.

# Results

## Structural impact of the STAT5B[Y665F] and STAT5B[Y665H] missense mutations in silico

Tyrosine 665 is the second most abundant mutational target in the STAT5B SH2 domain (Fig 1A), with 53 blood cancer cases identified in the Munich Leukemia Laboratory database and 12 blood cancer cases reported in the COSMIC database (Table S1). Although the STAT5B[Y665F] mutation was reproducibly associated with blood cancers in both databases, the STAT5B[Y665H] mutation was not found associated with cancer in either database indicating its presence is far less common than the STAT5B[Y665F] mutation. These mutations are especially interesting because STAT5B[Y665] is highly conserved across vertebrate species (Fig S1). To directly test the impact of the STAT5B[Y665F] and STAT5B[Y665H] mutations, we conducted comprehensive in silico analyses, combining structural prediction and pathogenicity assessment tools. Using AlphaFold3, we generated structures of the STAT5A and STAT5B SH2 domain homodimers. As expected from their

highly conserved sequences, STAT5A and STAT5B produced nearly identical structure predictions by AlphaFold3 (Abramson et al, 2024) (Fig S2A–D). The AlphaFold3-predicted structure of the STAT5B SH2 domain homodimer revealed that STAT5B[Y665] is located at a critical interface involved in STAT5B homodimerization (Fig 1B). This residue position and its surrounding structural elements are highly similar to what is predicted in the model of the STAT5A dimer interface generated by Fahrenkamp et al (2016).

We predicted the energetic contribution of each residue in the structures of both STAT5A and STAT5B homodimers using CO-ORDinator that is a neural network trained to take as input a protein backbone structure and output a sequence compatible with that backbone (Li et al, 2023). The version of COORDinator that we used was fine-tuned to predict the effects of amino acid substitutions on protein stability (Tsuboyama et al, 2023) using only the backbone structure as input. To distinguish energetic contributions specific to dimerization from more general contributions to domain stability, we compared the predicted energies of the STAT5B and STAT5A SH2 domain residues with and without their homodimeric counterparts (Fig S3A and B). This approach highlighted key residues within the C-terminal tail that form the antiparallel dimeric interface, as well as amino acids C-terminal to the phosphorylated Y699 in STAT5B (Fig 1C and D). We also used COORDinator to predict the energies of intramolecular interactions between the STAT5B C-terminal tail and the SH2 domain (Fig S3C). This approach highlighted STAT5B[F711] as making the greatest contribution to stabilizing the intramolecular interactions that support the dimer conformation (Fig 1E and F). Most substitutions to STAT5B[F711] and interacting residues such as STAT5B[Y665] were predicted to destabilize the intramolecular interaction (Fig 1G). Importantly, the STAT5B[Y665H] substitution introduces an imidazole and was predicted by COORDinator to destabilize binding of the C-terminal tail (Fig 1G). In contrast, STAT5B[Y665F] was predicted to stabilize the structure, perhaps by promoting intramolecular aromatic stacking interactions with STAT5B[F711].

To further evaluate the potential pathogenicity of these two mutations, we employed multiple state-of-the-art computational prediction tools. AlphaMissense (Cheng et al, 2023) predicted mild functional impact for both mutations with scores of 0.173 and 0.383 for STAT5B[Y665F] and STAT5B[Y665H], respectively, categorizing them as benign. However, the Combined Annotation Dependent Depletion (CADD) (Rentzsch et al, 2019) PHRED scores of 24.3 (STAT5B[Y665F]) and 23.1 (STAT5B[Y665H]) suggested potential deleterious effects, as scores above 20 are often considered impactful. Rare Exome Variant Ensemble Learner (REVEL) (Ioannidis et al, 2016) analysis yielded scores of 0.535 for STAT5B[Y665F] and 0.304 for STAT5B[Y665H], indicating a higher probability of pathogenicity for STAT5B[Y665F]. Notably, PolyPhen-2 (Adzhubei et al, 2013) showed marked differences between the mutations, with STAT5B[Y665F] scoring 0.93 (probably damaging) and STAT5B[Y665H] scoring 0.084 (benign). In summary, different tools provided a range of pathophysiological predictions for the mutants. Only COORDinator, the only tool that uses structural modeling to predict energetic effects rather than pathogenicity, gave predictions that fully match the experimental results.

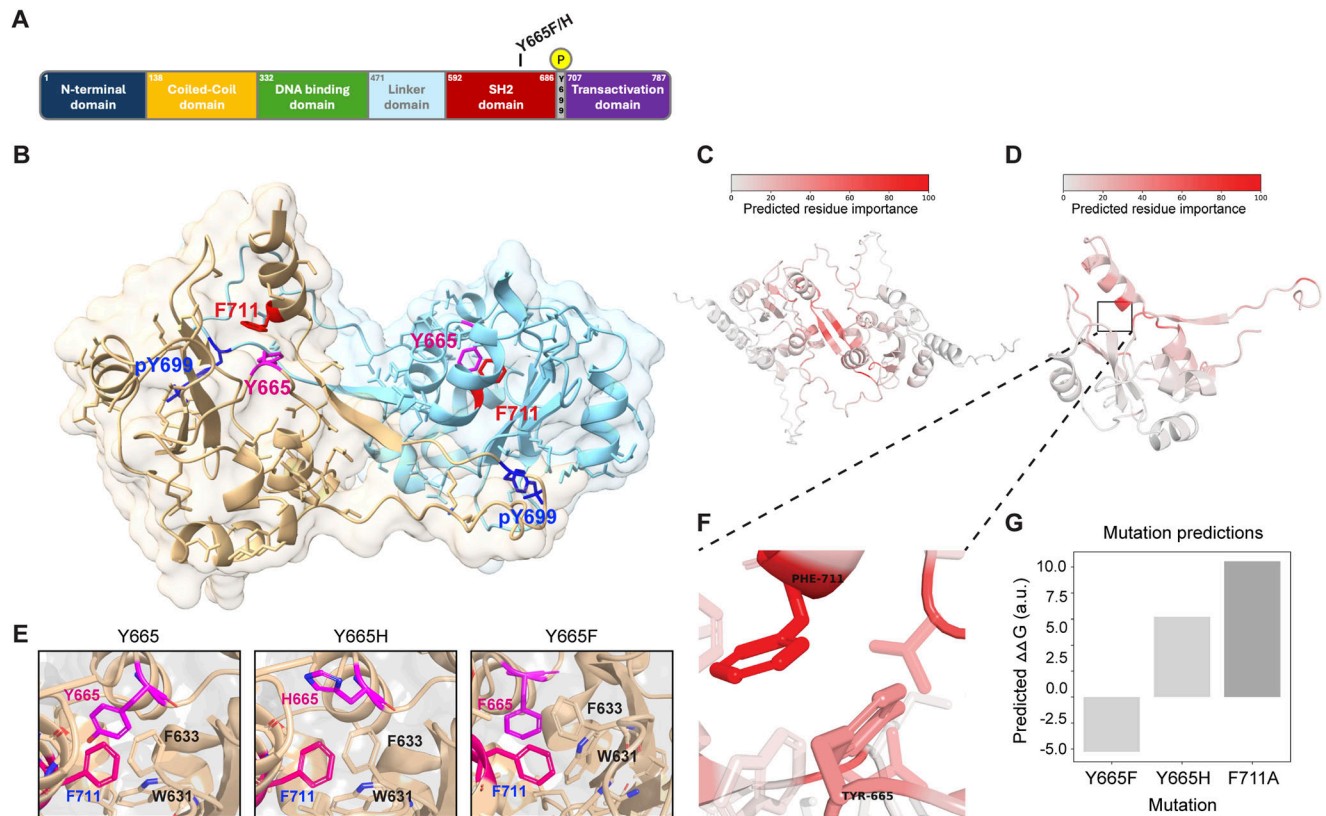

**Figure 1. STAT5B SH2 dimerization modeled by AlphaFold3.**
**(A)** Schematic of human STAT5B protein domains showing the locations of tyrosine 665 (Y665) and phospho-tyrosine 699 (pY699). **(B)** Structure of the human STAT5B SH2 homodimer generated by AlphaFold3. Binding pockets of key residues pY699 (blue) and F711 (red) are indicated. The STAT5B model shows the hydrophobic binding pocket containing the key residue Y665 (magenta). **(C)** Structure of the human STAT5B SH2 homodimer generated by AlphaFold3 with residues colored red relative to their importance to the binding interface as predicted by COORDinator. **(D)** Structure of the human STAT5B SH2 monomer in its dimeric conformation generated by AlphaFold3 with residues colored red relative to their importance to stabilization of the C-terminal tail as predicted by COORDinator. **(E)** Model of the human STAT5B SH2 homodimer with tyrosine, histidine, or phenylalanine at position 665, as predicted by AlphaFold3. **(F)** STAT5B model highlights the intramolecular interaction between F711 and the hydrophobic binding pocket containing the key residue Y665. **(G)** Comparison of the predicted energetic consequences of the F711A, Y665F, and Y665H mutations on stabilization of the C-terminal tail. Relative mutational effects are annotated using ΔΔG, in arbitrary units; stabilizing mutations have a negative value, and destabilizing mutations have a positive value.

## STAT5B[Y665F] but not STAT5B[Y665H] activates genetic programs in CD4[+] T cells in vitro

To assess the impact of the STAT5B[Y665F] and STAT5B[Y665H] mutations on the transcriptional activity of STAT5B, we measured gene expression in $Stat5a/b^{-/-}$ T cells reconstituted with either WT or mutant STAT5B. Naïve T cells were purified from lymph nodes and spleens of WT or $Stat5a/b^{-/-}$ mice (Cui et al, 2004), activated in vitro, and transduced with retroviral vectors encoding WT STAT5B, STAT5B[Y665F], and STAT5B[Y665H] (Fig 2A). STAT5B[N642H], an established activating mutation identified in T-cell leukemia patients (de Araujo et al, 2019), was included as a positive control. We confirmed an ~98% reduction of $Stat5b$ mRNA in $Stat5a/b^{-/-}$ CD4[+] cells as compared to WT cells using RNA-sequencing (RNA-seq) data, demonstrating efficient deletion of the $Stat5a/b$ locus and providing a null background for testing the different mutants (Fig 2B). Upon retroviral introduction of the various constructs, $Stat5b$ mRNA levels were restored to near WT levels in the $Stat5a/b^{-/-}$ cells. Notably, both positive control STAT5B[N642H] and experimental

STAT5B[Y665F] induced surface expression of the two well-known STAT5 target genes $CD25$ ($IL2r\alpha$) and CD98 (Villarino et al, 2022), whereas STAT5B[Y665H] failed to do so (Fig 2C). Transcriptome analysis showed that WT STAT5B mobilized 141 differentially expressed genes (DEGs) versus "empty" retrovirus, whereas positive control STAT5B[N642H] and experimental STAT5B[Y665F] and STAT5B[Y665H] induced 363, 318, and 23 genes, respectively (Table S2). Of the 318 genes mobilized by STAT5B[Y665F], about one third (113 genes) are directly engaged by STAT5B as determined by ChIP-seq and are therefore considered bona fide STAT5B target genes (Table S2). Many DEGs (287) were shared between the positive control STAT5B[N642H] and experimental STAT5B[Y665F] mutants, indicating that, overall, these variants demonstrated similar transcriptional effects (Fig 2D; Table S2). Common bona fide STAT5 target genes, such as $Socs2$ and $Cish$, were activated up to several thousand-fold, whereas more cell-restricted genes, such as lymphotoxin alpha ($Lta$), $Irf8$, and the interleukin-2 receptor alpha ($Il2ra$), were induced up to 10-fold (Fig 2E–G). In contrast, STAT5B[Y665H] failed to activate gene expression in the $Stat5a/b^{-/-}$ background. These results

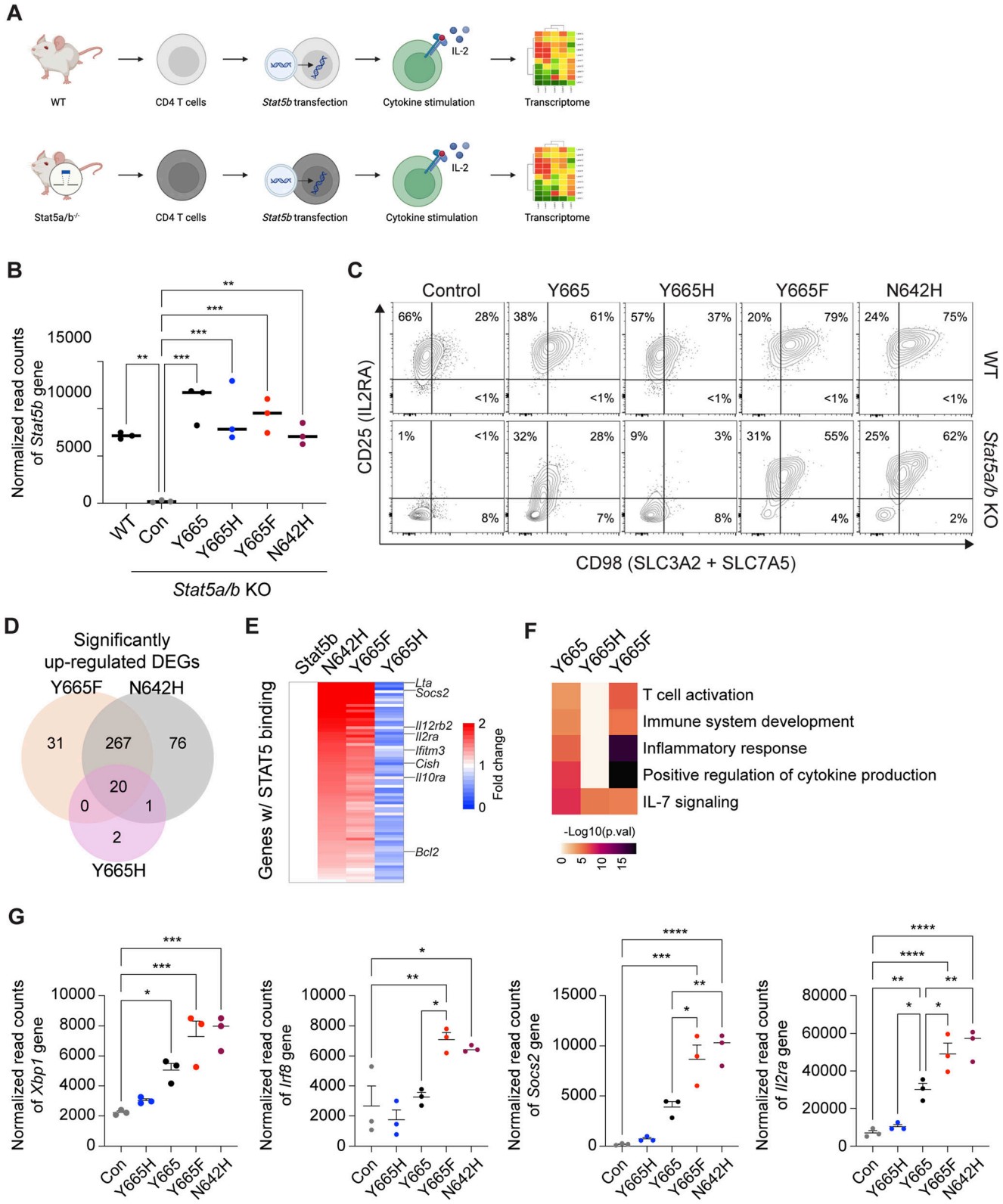

**Figure 2. Transcriptomes activated by WT and STAT5B mutations in *Stat5a/b*-null T cells.**
**(A)** Schematic of the experimental approach to assess the impact of STAT5B[Y665] mutations on naïve CD4 T cells from lymph nodes and spleens of WT or STAT5A/B-deficient mice. **(B)** Normalized read counts (TPM, tags per million) of the *Stat5b* gene in WT and *Stat5a/b* KO T cells transduced with a control (Con) retroviral vector or vectors encoding STAT5B, WT Y665, Y665F, Y665H, and N624H, evaluated by RNA-seq. **(C)** Representative flow cytometry contour plots showing CD4⁺ cell population (*n* = 3). **(D)** Venn diagram displaying the number of significantly induced genes by retroviral vectors encoding Y665F, Y665H, and N642H mutations compared with WT evaluated

demonstrate that STAT5B$^{Y665F}$ and STAT5B$^{N642H}$ have equivalent GOF capacity, whereas STAT5B$^{Y665H}$ is a clear LOF mutation.

## Introduction of *Stat5b*$^{Y665F}$ and *Stat5b*$^{Y665H}$ into the mouse genome

To investigate in vivo consequences of the STAT5B$^{Y665F}$ and STAT5B$^{Y665H}$ mutations, we introduced them into mouse genome using CRISPR/Cas9 and base editing (see the Materials and Methods section for details). Of note, although the expected 25% homozygous mating was obtained from *Stat5b*$^{Y665H}$ mating, only 13% homozygosity was obtained by breeding heterozygous *Stat5b*$^{Y665F}$ mice (Lee et al, 2025). The reason for the loss of ~50% of the homozygous mice is not known. Both ex vivo and in vivo studies were performed (Fig 3A). Although homozygous *Stat5b*$^{Y665F}$ and *Stat5b*$^{Y665H}$ mice displayed an overall normal appearance at the time of weaning, the latter had a retarded body growth, consistent with suboptimal growth hormone signaling in the presence of a LOF Stat5b mutation. An especially progressive and morbid form of ulcerative dermatitis, reminiscent of the type known to occur in the inbred C57BL/6 strain, developed in the *Stat5b*$^{Y665F}$ mice necessitating humane euthanasia of both heterozygous and homozygous mice at early ages. Notably, the condition developed at a significantly earlier age in the homozygous as compared to heterozygous mice (Fig 3B). In contrast, homozygous *Stat5b*$^{Y665H}$ mice did not develop either dermatitis or any other condition requiring early euthanasia over the first year of life. Both heterozygous and homozygous *Stat5b*$^{Y665F}$ mice demonstrated splenomegaly as compared to WT mice, whereas significantly smaller spleens were observed in homozygous *Stat5b*$^{Y665H}$ mice (Fig 3C).

## Increased CD8$^+$ central memory cells and CD4$^+$ Tregs in splenocytes from *Stat5b*$^{Y665F}$ mice

To address the question of which cell populations were present in the differentially sized spleens of the *Stat5b*$^{Y665F}$ and *Stat5b*$^{Y665H}$ mice, we quantified splenic immune cell populations, focusing on effector, regulatory, and memory T cells using flow cytometry (Figs 3D–G and S4B). Homozygous *Stat5b*$^{Y665F}$ mice exhibited a significantly increased total splenocyte number, whereas *Stat5b*$^{Y665H}$ mice displayed a significant reduction in cell numbers as compared to WT controls (Fig 3D). Total CD8$^+$ cells were expanded in *Stat5b*$^{Y665F}$ mice but significantly reduced in *Stat5b*$^{Y665H}$ mice (Figs 3E and S4B). No significant differences were observed in the cell numbers of CD4$^+$ and neutrophil populations, and natural killer (NK) cells in the two mutants as compared to WT controls. Central and effector memory cells, crucial for long-term immunological memory, immune surveillance, and response within peripheral tissues, were significantly and selectively elevated within the CD8$^+$ T-cell compartment in the *Stat5b*$^{Y665F}$ mice (Figs 3F and S4B). In contrast, these

findings were not present in the CD4$^+$ T-cell compartment. Regulatory T cells (Tregs) showed extensive expansion in *Stat5b*$^{Y665F}$ mice (Figs 3G and S4B), whereas reduced STAT5B function in *Stat5b*$^{Y665H}$ mice was not associated with significant changes in Treg numbers. We then investigated whether or not similar changes in T-cell compartments could be identified in peripheral blood. Similar to the spleen, we noted a significant increase in lymphocytes, and CD8$^+$ and CD4$^+$ T cells in 2-mo-old homozygous *Stat5b*$^{Y665F}$ mice and a significant decrease in *Stat5b*$^{Y665H}$ mice (Fig 3H–J). To test whether or not these differences were stable with age, repeat assays were performed on 11-mo-old mice. At that timepoint, because of the necessity for humane euthanasia at earlier ages in the *Stat5b*$^{Y665F}$ homozygous mice, only heterozygous *Stat5b*$^{Y665F}$ mice were available for comparison with the homozygous *Stat5b*$^{Y665H}$ mice. Findings in older heterozygous *Stat5b*$^{Y665F}$ mice resembled those found in younger homozygous *Stat5b*$^{Y665F}$ mice, and no significant differences between younger and older *Stat5b*$^{Y665H}$ mice were found (Figs S5A–G and S6A–J). No signs of leukemia were observed in homozygous *Stat5b*$^{Y665F}$ mice at 2 mo of age (the oldest age cohorts of mice could be followed before the development of complicating dermatitis), and heterozygous *Stat5b*$^{Y665F}$ and homozygous *Stat5b*$^{Y665H}$ mice by 11 mo of age.

## Expansion of central memory T cells in *Stat5b*$^{Y665F}$ mice

To further investigate the effects of mutant Stat5 on gene expression in T cells, we isolated CD45$^+$ cells from spleens of WT and mutant mice and performed single-cell RNA sequencing (scRNA-seq). CD45$^+$ is a major transmembrane glycoprotein expressed on all nucleated hematopoietic cell and a well-established white blood cell marker. After identifying key T-cell populations (Fig 4A), we noted that total CD8$^+$ cells were expanded in *Stat5b*$^{Y665F}$ mice and reduced in *Stat5b*$^{Y665H}$ mice, as were B cells and monocytes (Fig 4B). Central memory T cells (Tcm) were significantly expanded in Stat5b$^{Y665F}$ mice compared with WT or Stat5b$^{Y665H}$ mice, similar to the flow cytometry results (Fig 4C). Known Stat5-regulated genes were also enriched in *Stat5b*$^{Y665F}$ Tcm including *Gzmb*, *Cish*, and *Il12rb* (Fig 4D). The scRNA-seq also revealed an expansion of CD4$^+$ T$_{regs}$ exclusively in *Stat5b*$^{Y665F}$ mice (Fig 4E), with the expression of Stat5-dependent genes associated with cellular homeostasis, Th17 cell differentiation, and inflammatory responses in these cells (Fig 4F and G). These findings highlight the distinct roles of activating and inactivating STAT5B mutations in shaping T-cell subtypes and their functional states.

## STAT5B$^{Y665F}$ demonstrates enhanced STAT5B$^{Y699}$ tyrosine phosphorylation upon interleukin stimulation

Phosphorylation of tyrosine 694 in STAT5A and tyrosine 699 in STAT5B is required for transcriptional capacity and immune cell regulation (Lin et al, 2024). To gauge the impact of the two STAT5B

---

using RNA-seq. Cells were stimulated with IL-2 (*n* = 3). **(E)** Heatmaps showing fold changes of significantly up-regulated genes between N642H, Y665F, and Y665H on Stat5b (Y665) as 1 in STAT5A/B-deficient T cells. **(F)** Heatmap of genes expressed at significantly higher levels in each sample and significantly enriched in Gene Ontology (GO) terms. **(G)** Dot plots of the normalized read counts (TPM) for mRNA levels of five genes regulated by STAT5B. Results are shown as the means ± SEM of independent biological replicates. *P*-values are from one-way ANOVA with Tukey's multiple comparisons test. *$P$ < 0.05, **$P$ < 0.01, ***$P$ < 0.0001, ****$P$ < 0.0001.

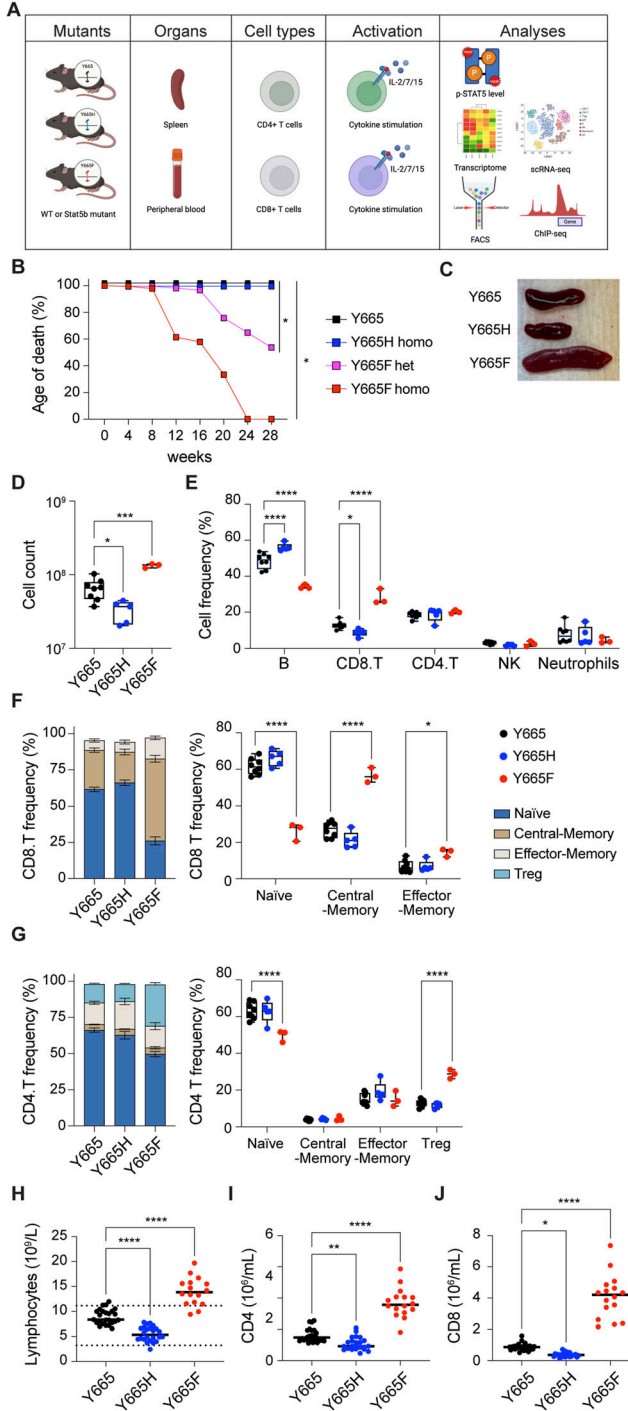

**Figure 3. Altered T-cell distribution in spleens of *Stat5b* mutant mice.**
**(A)** Schematic illustration of the experimental approach. The *Stat5b*[Y665] variants were introduced into the mouse genome, and T cells and stem cells from immune system organs. Intact or cytokine-stimulated cells were subjected to Western blot, RNA-seq, scRNA-seq, FACS, and ChIP-seq analyses. **(B)** Kaplan–Meier curves exhibiting the age of death. Y665, *n* = 101; Y665H, *n* = 187; Y665F[+/−], *n* = 217; Y665F[−/−], *n* = 46. **(C)** Images of spleens from WT and mutant mice. **(D)** Box plots showing the total cell number in spleen from WT and mutant mice analyzed via flow cytometry (Y665, *n* = 8; Y665H, *n* = 5; Y665F, *n* = 3). *P*-values are from one-way ANOVA with Tukey's multiple comparisons test. Median, middle bar inside the box; IQR, 50% of the data; whiskers, 1.5 times the IQR. **(E)** Percentages of B cells, CD4 T cells, CD8 T cells, NK cells, and neutrophils were calculated using FACS. **(F, G)**

mutations, we cultured spleen-derived primary T cells from WT and mutant mice with IL-2, IL-7, and IL-15, and measured phospho-STAT5B[Y699] (pY-STAT5B) by flow cytometry (Fig 5A and B). All three interleukins induced STAT5B[Y699] tyrosine phosphorylation in the STAT5B[Y665F] but not the STAT5B[Y665H] mutant CD4[+] and CD8[+] T cells. Importantly, STAT5B[Y699] phosphorylation levels were significantly higher in stimulated *Stat5b*[Y665F] mutant cells compared with stimulated WT controls. This difference was most prominent in CD4[+] cells, where pY-STAT5B levels in *Stat5b*[Y665F] mutant cells exceeded WT levels by 5–10-fold in the total CD4[+] and CD8[+] T-cell populations (Fig 5B). Relative basal expression levels of components of the respective receptors for IL-2, IL-7, and IL-15 in spleen-derived T cells without stimulation from *Stat5b*[Y665], *Stat5b*[Y665H], and *Stat5b*[Y665F] mice demonstrated that expression levels of *Il2ra* (CD25) were significantly lower in the *Stat5b*[Y665] and *Stat5b*[Y665H] T cells as compared to *Stat5b*[Y665F] mice (Fig 5C). The higher expression levels of *Il2ra* in the *Stat5b*[Y665F] mice could be considered secondary to both the lower proportion of naïve T cells in this population (Fig 3F and G) and the proposed enhanced dimer stabilization mediated by the substitution of phenylalanine for the tyrosine residue (Fig 1G), both of which would be hypothesized to occur if *Stat5b*[Y665F] represents a GOF mutant. In contrast to the differences shown in the *Stat5b*[Y665F] mice, the relative proportions of naïve T cells in T-cell populations derived from *Stat5b*[Y665] WT and *Stat5b*[Y665H] mutant mice were relatively similar (Fig 3F and G), but expression levels of *Il2ra* and *Il7r* were significantly lower in the *Stat5b*[Y665H] as compared to the *Stat5b*[Y665] mice. Both the lower expression levels and the absence of significant STAT5 phosphorylation after cytokine stimulation are consistent with the notion that STAT5B[Y665H] could represent a LOF mutant.

## Opposing impact of STAT5B[Y665F] and STAT5B[Y665H] on transcriptional programs and enhancer establishment

To gain a more comprehensive understanding of the STAT5B[Y665F] and STAT5B[Y665H] mutants' capacity to engage with the genome and activate biological programs, we conducted transcriptomic and ChIP-seq studies. A caveat of these studies is that proportions of T-cell subsets were altered by the expression of either the inhibitory STAT5B[Y665H] or activating STAT5B[Y665F] mutants. Most remarkably, there was the relative increase in central memory and the relative decrease in naïve CD8[+] T cells in STAT5B[Y665F] as compared to STAT5B[Y665] and STAT5B[Y665H] mice. This is compatible with the notion that some of the observed differences could be explained by differences in T-cell subset proportions rather than being directly attributed to differences in binding affinity. At the same time, subset differences were less marked between STAT5B[Y665] and STAT5B[Y665H] mice; therefore, alterations found there

Percentages of CD8 and CD4 subpopulations. *P*-values are from two-way ANOVA with Tukey's multiple comparisons test. *$P < 0.05$, **$P < 0.01$, ***$P < 0.0001$, ****$P < 0.0001$. **(H)** Total lymphocyte counts in peripheral blood from 7 to 10-wk-old adult WT and mutant mice. Results are shown as the median of independent biological replicates (Y665, *n* = 25; Y665H, *n* = 24; Y665F, *n* = 16). **(I, J)** Numbers of CD4[+] and CD8[+] T cells identified by flow cytometry. Results are shown as the median of independent biological replicates. Statistical significance was assessed using one-way ANOVA followed by Tukey's multiple comparisons test.

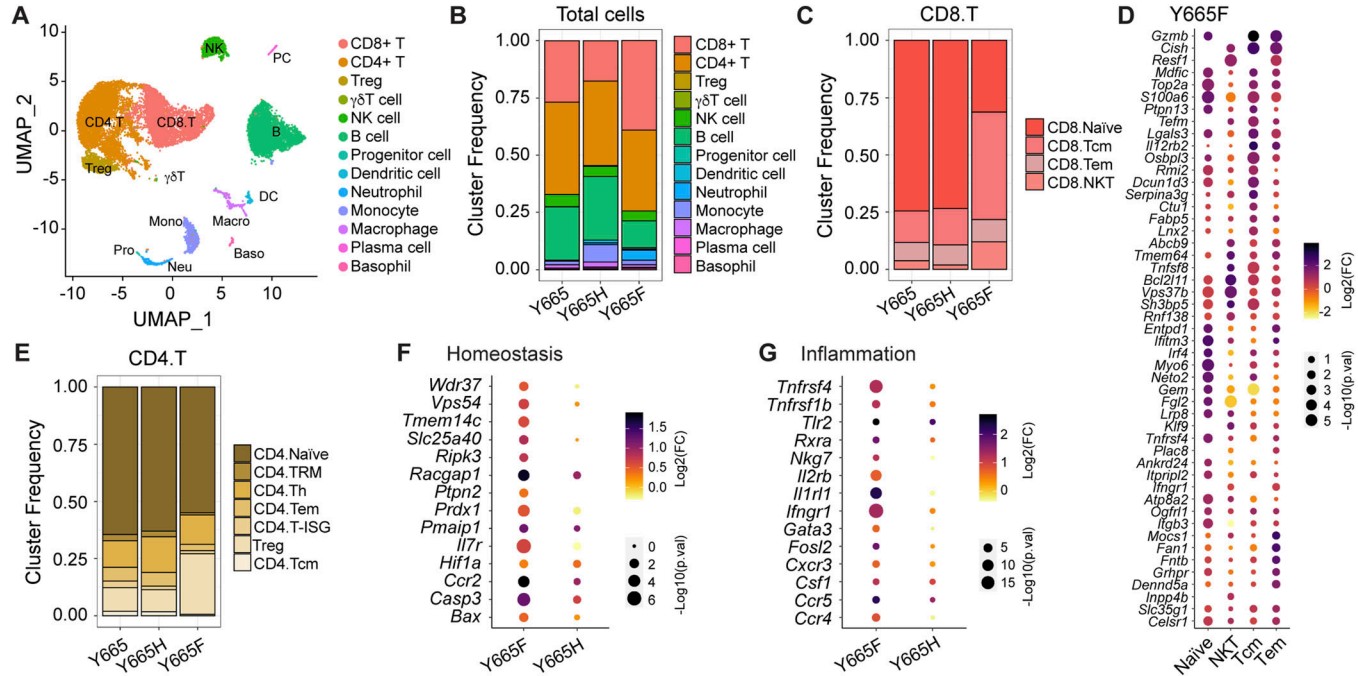

**Figure 4. Immune dynamics in spleen analyzed using scRNA-seq.**
**(A)** Uniform Manifold Approximation and Projection graph of scRNA-seq data from lymph nodes of WT and mutant mice, colored by cell population (*n* = 1 combined sample from three mice of each group). **(B)** Heatmap of cell population. **(C)** Frequencies of CD8 subtypes: Tcm (central memory T cells), Tem (effector memory T cells), and NKT (NK-like T cells). **(D)** Dot plot of differentially expressed genes related to cytokine signaling and T-cell activation in CD8 T-cell subsets. **(E)** Frequencies of CD4 subtypes: TRM (tissue-resident memory T cells), Th (helper T cells), and Treg (regulatory T cells). **(F, G)** Categorical scatter plot of relative gene expression values of differentially expressed genes related to homeostasis (F) and inflammatory response (G) in regulatory CD4 T-cell subpopulations.

may have a stronger relationship with binding affinity. The data should be viewed as a representative of a total T-cell population composed of different proportions of T-cell subsets. We performed "bulk" RNA-seq on splenic T cells isolated from *Stat5b*^Y665F and WT mice after IL-2/IL-7 stimulation. 10,010 up-regulated DEGs with STAT5 binding on their regulatory elements were identified in *Stat5b*^Y665F mice relative to WT controls, with particular enrichment in immune system processes and activation (Fig 5D–F; Table S3). These ex vivo experiments demonstrated the enhanced capacity of the STAT5B^Y665F mutant to activate bona fide STAT5 target genes, but this did not provide information on impact of the mutation on the steady-state physiology of immune programs in vivo. To address the impact of the two mutants on steady-state programs, we conducted bulk RNA-seq on T cells immediately upon their isolation, that is, without additional interleukin stimulation. The expression of ~150 genes was elevated in *Stat5b*^Y665F splenocytes as compared to *Stat5b*^Y665H cells (Table S4). The STAT5-bound genes were enriched in T-cell activation and inflammatory response genes including genes typically induced by IFNs, compatible with the changes in T-cell subsets observed in the different mutants (Fig 5G–J).

To identify genes that are directly impacted by WT and the two mutant STAT5B proteins, we conducted ChIP-seq on splenic T cells stimulated with IL-2 and IL-7, and measured STAT5B, PolII, and H3K27ac occupancy. The data permitted the identification of candidate enhancers, both globally and at specific immune-regulated loci (Fig 6A–C). Approximately 4,400 STAT5B binding peaks

coinciding with H3K27ac marks were identified in *Stat5b*^Y665F cells, whereas strikingly few sites bound by STAT5B were detected in *Stat5b*^Y665H cells, in agreement with the in silico predictions presented above that the STAT5B^Y665H variant would exhibit weakened dimerization strength through destabilization of the intramolecular binding with the C-terminal tail. Weakened dimerization strength would then compromise the ability of the STAT5B^Y665H variant to access the genome. In *Stat5b*^Y665F cells, STAT5B binding associated with H3K27ac marks at putative distal enhancers (1,465 genes), promoter regions (636 genes), intronic sequences (2,097 genes), and 3′UTR (229 genes) (Fig 6B; Table S5). Representative ChIP-seq data visualizing candidate enhancers located in the 5′ upstream region (*Cd8a*), promoter sequences (*Il18r1*), within introns (*Il2rb*), and 3′ flanking regions (*Il2rb*) are shown in Fig 6D. We also identified 155 STAT5B-bound enhancer clusters in Stat5b^Y665F cells (Fig 6C; Table S5), also referred to as super-enhancers or stretch enhancers. These enhancer clusters have been previously associated with pan-lineage STAT5 target genes, such as *Cish* and *Socs1*, and with lineage-specific genes, such as the *Il2ra* (Li et al, 2017), the *Osm-Lif* locus, and more cell-restricted loci (Shin et al, 2016; Lee et al, 2021; Lee et al, 2023). STAT5B binding was detected in the proximity of about 10% of the DEG in unstimulated splenic T cells and in approximately one half of the genes induced in primary splenic T cells cultured in the presence of interleukins (Table S4), suggesting that they are direct transcriptional targets. This group includes genes encoding key immune receptors, including *Il2ra* and *Il12rb2*, and key TFs, including *Stat1*, *Irf1*, and *Irf8* (Table S4). These findings further

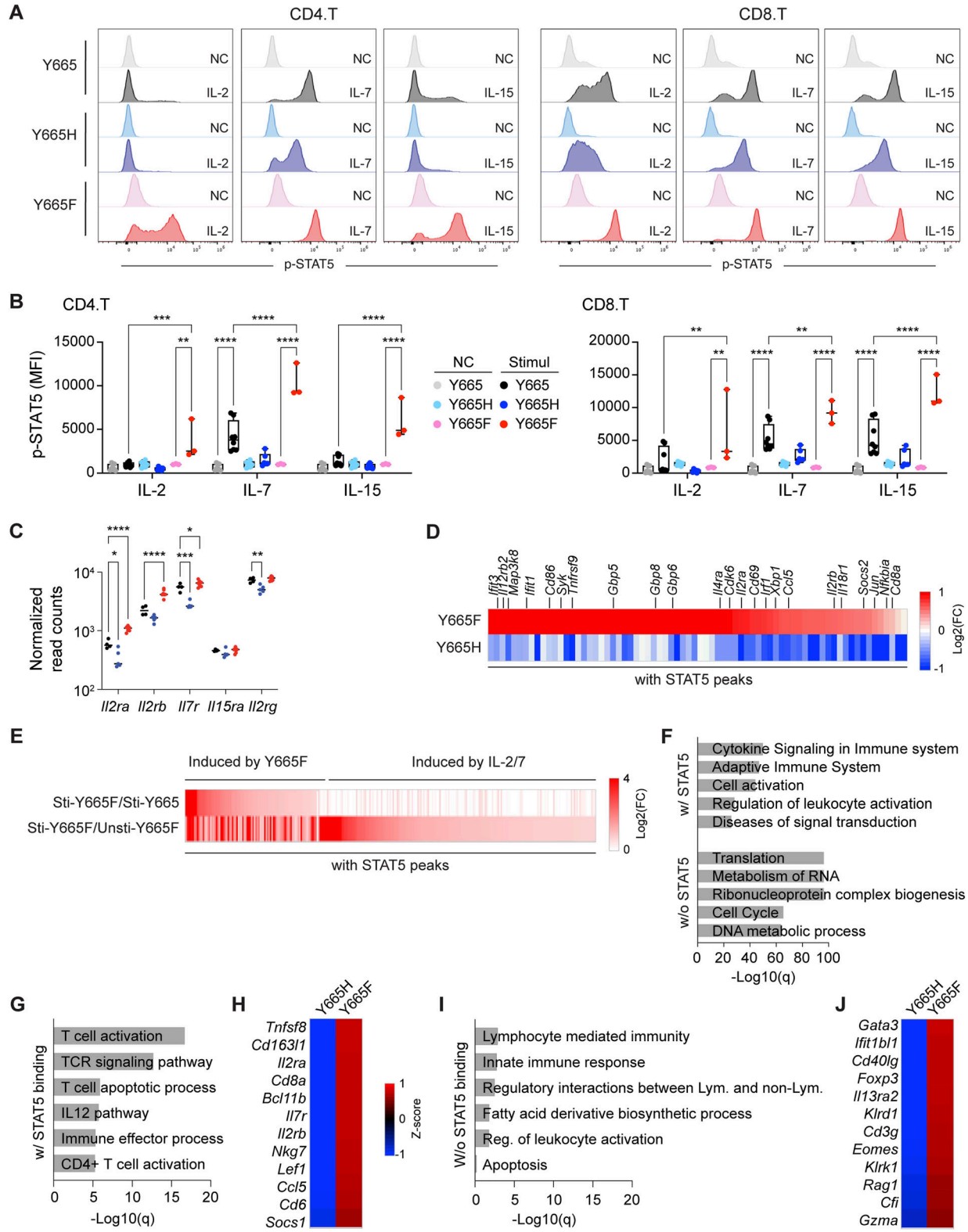

**Figure 5. Opposing transcriptional activity of STAT5B$^{Y665F}$ and STAT5B$^{Y665H}$ at steady-state levels.**
**(A)** Representative histograms of the flow cytometry analyses (FACS) of phosphorylated STAT5 (pY-STAT5) in CD4$^+$ and CD8$^+$ T cells from *Stat5b*$^{Y665}$ (Y665), and *Stat5b*$^{Y665H}$ (Y665H) and *Stat5b*$^{Y665F}$ (Y665F) mice after interleukin (IL)-2, IL-7, and IL-15 in vitro stimulation compared with no cytokine (NC) controls. **(B)** Dot blot graphs showing quantification of phospho-STAT5 levels in CD4$^+$ and CD8$^+$ T cells from multiple mice of each genotype from FACS (Y665, *n* = 8; Y665H, *n* = 6; Y665F, *n* = 3). *MFI*, median fluorescence intensity. *P*-values are derived from two-way ANOVA with Tukey's multiple comparisons test. **P < 0.01, ***P < 0.0001, ****P < 0.0001. **(C)** Normalized read

emphasize that STAT5B$^{Y665F}$ is a GOF mutation and STAT5B$^{Y665H}$ is largely inactive.

### Outcome of STAT5B$^{Y665F}$ and STAT5B$^{Y665H}$ variants on hematopoietic stem and progenitor cells in bone marrow

Because another leukemia-associated activating STAT5B mutation, STAT5B$^{N642H}$, produces an expansion of hematopoietic stem cells (HSCs) in the bone marrow (BM) when expressed as a *Vav-1*–driven transgene in mice (Pham et al, 2018), we performed a comparative analysis of the impact of the *Stat5b*$^{Y665F}$ and *Stat5b*$^{Y665H}$ mutations on early hematopoiesis in the BM (Fig 7). The total number of BM cells was not significantly different between the *Stat5b*$^{Y665}$, *Stat5b*$^{Y665F}$, and *Stat5b*$^{Y665H}$ mice (Fig 7A). Homozygous *Stat5b*$^{Y665F}$ mice showed a significantly higher proportion and a total number of KSL (Lin-Sca-1$^+$CD117$^+$) cells (Fig 7B and C), as well as subset populations including long-term hematopoietic stem cells (LT-HSCs), multipotent progenitor (MPP) 2 cells and MPP3&4, relative to homozygous *Stat5b*$^{Y665}$ and *Stat5b*$^{Y665H}$ mice (Fig 7D–F). BM cells from homozygous *Stat5b*$^{Y665F}$ mice demonstrated a higher frequency of colony-forming cells at day 10 (Fig 7G), indicating that the expanded KSL cells in F665 mutant mice were functional. There were no significant differences in myeloid progenitor (MP) cell (Fig 7H) and subset populations including common myeloid progenitors (Fig 7I), granulocyte–monocyte progenitors (Fig 7J), and megakaryocytic erythroid progenitors between *Stat5b*$^{Y665}$, *Stat5b*$^{Y665F}$, and *Stat5b*$^{Y665H}$ mice (Fig 7K). There also were no significant differences in numbers of common lymphoid progenitors (Fig 7L). A Uniform Manifold Approximation and Projection graph shows an overview of the cell types identified from all three genotypes using single-cell sequencing (Fig 7M). The *Stat5b*$^{Y665F}$ mice showed a striking expansion of LT-HSCs in comparison with the other two genotypes (Fig 7N). Single-cell sequencing revealed that *Stat5b*$^{Y665F}$ mice demonstrated higher expression levels of genes associated with the Gene Ontology (GO) category "hematopoiesis stem cells" in their LT-HSCs as compared to *Stat5b*$^{Y665H}$ mice (Fig 7O). Overall, results showed an expansion of HSCs in the *Stat5b*$^{Y665F}$ mice.

## Discussion

The primary objective of this study was to understand the structural, genomic, and pathophysiological impacts of two human leukemic STAT5B mutations (Rajala et al, 2013; Kiel et al, 2014), STAT5B$^{Y665F}$ and STAT5B$^{Y665H}$, identified in patients with T-cell leukemias (Rajala et al, 2013; Kiel et al, 2014). For this, we combined

comprehensive in silico analyses, primary cell studies, and experimental mouse genetics.

The complementary in silico approaches, AlphaFold3, COORDinator, AlphaMissense, and the pathogenicity prediction databases CADD (Rentzsch et al, 2019), REVEL (Ioannidis et al, 2016), and PolyPhen-2 (Adzhubei et al, 2013) presented a complex picture of the mutations' potential effects. Our structural analyses suggested both mutations could impact STAT5B dimerization. However, the pathogenicity predictions were inconsistent highlighting the challenges in relying solely on computational methods for assessing the impact of specific mutations.

In vitro transcriptomic studies using primary T cells confirmed GOF properties for STAT5B$^{Y665F}$ and LOF for STAT5B$^{Y665H}$, consistent with COORDinator predictions and PolyPhen-2 analysis. Ex vivo and in vivo studies in mice carrying these mutations further established STAT5B$^{Y665F}$ as a GOF and STAT5B$^{Y665H}$ as a LOF mutation. On a molecular level, STAT5B$^{Y665F}$ exhibited an elevated capacity to establish genomic enhancers and hyperactivate transcriptional programs, leading to elevated CD8$^+$ T-cell populations. The elevation in Treg cells is compatible with previous observations of a permissive role of STAT5B in Treg development (Xiang et al, 2020). In contrast, STAT5B$^{Y665H}$ was predicted to have reduced dimerization strength and showed little binding to genomic enhancers and activation of interleukin-induced genetic programs, reminiscent of what has been observed in mice lacking Stat5b (Udy et al, 1997) and the STAT5B$^{Y699F}$ mutant that fails to dimerize (Lin et al, 2024).

The GOF properties of the STAT5B$^{Y665F}$ mutant validated in vivo are postulated to arise from an enhanced stabilization of intramolecular interactions between the STAT5B C-terminal tail and the SH2 domain through intramolecular aromatic stacking interactions with STAT5B$^{F711}$. This enhanced intramolecular stabilization would then stabilize STAT5B dimer formation, which would in turn lead to enhanced transcriptional activation. In WT STAT5B, dimer formation is known to be most profoundly stabilized by phosphorylation of tyrosine 699 (Lin et al, 2024). We conjecture here that the STAT5B$^{Y665F}$ mutation further stabilizes the dimer, thus leading to the enhanced gene transcription and downstream effects observed here. It has been reported that phosphorylation of additional tyrosine residues in the activation domain of STAT5B (Y725, Y740, and Y743) can influence its function either positively (Y725) or negatively (Y740, Y743) (Weaver & Silva, 2006); however, there are no reports indicating that phosphorylation of Y665 is critical for STAT5B function. Therefore, we have no evidence to suggest that the F residue mutation mimics phosphorylation of tyrosine in this case. Instead, we consider that the mechanism of the enhanced transcription in the STAT5B$^{Y665F}$ mutant is the positive energetic contribution to the intramolecular interaction between the C-terminal

---

counts (TPM, tags per million) of interleukin-2 receptor subunit alpha (*Il2ra*), interleukin-2 receptor subunit beta (*Il2rb*), interleukin-17 receptor A (*Il17r*), interleukin-15 receptor subunit alpha (*Il15ra*), and interleukin-2 receptor subunit gamma (Il2rg) expression levels evaluated by RNA-seq in T cells from *Stat5b*$^{Y665}$ (Y665), *Stat5b*$^{Y665H}$ (Y665H), and *Stat5b*$^{Y665F}$ (Y665F) mice in the absence of in vitro cytokine stimulation. **(D)** Heatmaps depicting fold changes of significantly differentially regulated genes from *Stat5b*$^{Y665F}$ and *Stat5b*$^{Y665H}$ mice. **(E)** Heatmaps depicting fold changes of significantly differentially up-regulated and enriched genes that demonstrate STAT5 binding on their regulatory elements from unstimulated or stimulated T cells from *Stat5b*$^{Y665}$ WT and *Stat5b*$^{Y665F}$ mutant mice (IL-2/7–stimulated Y665, n = 4; IL-2/7–stimulated Y665F, n = 3). **(F)** Gene categories expressed at significantly higher levels from genes that demonstrate STAT5 binding on their regulatory elements in stimulated T cells from *Stat5b*$^{Y665F}$ mice as compared to *Stat5b*$^{Y665}$ mice. **(G, H)** Gene categories expressed at significantly higher levels from genes that demonstrate STAT5 binding on their regulatory elements in *Stat5b*$^{Y665H}$ mice as compared to *Stat5b*$^{Y665F}$ mice. **(I, J)** Gene categories significantly differentially up-regulated and enriched genes related to T-cell activation and apoptosis in *STAT5B*$^{Y665H}$ as compared to *STAT5B*$^{Y665F}$ mice (Y665H, n = 5; Y665F, n = 7).

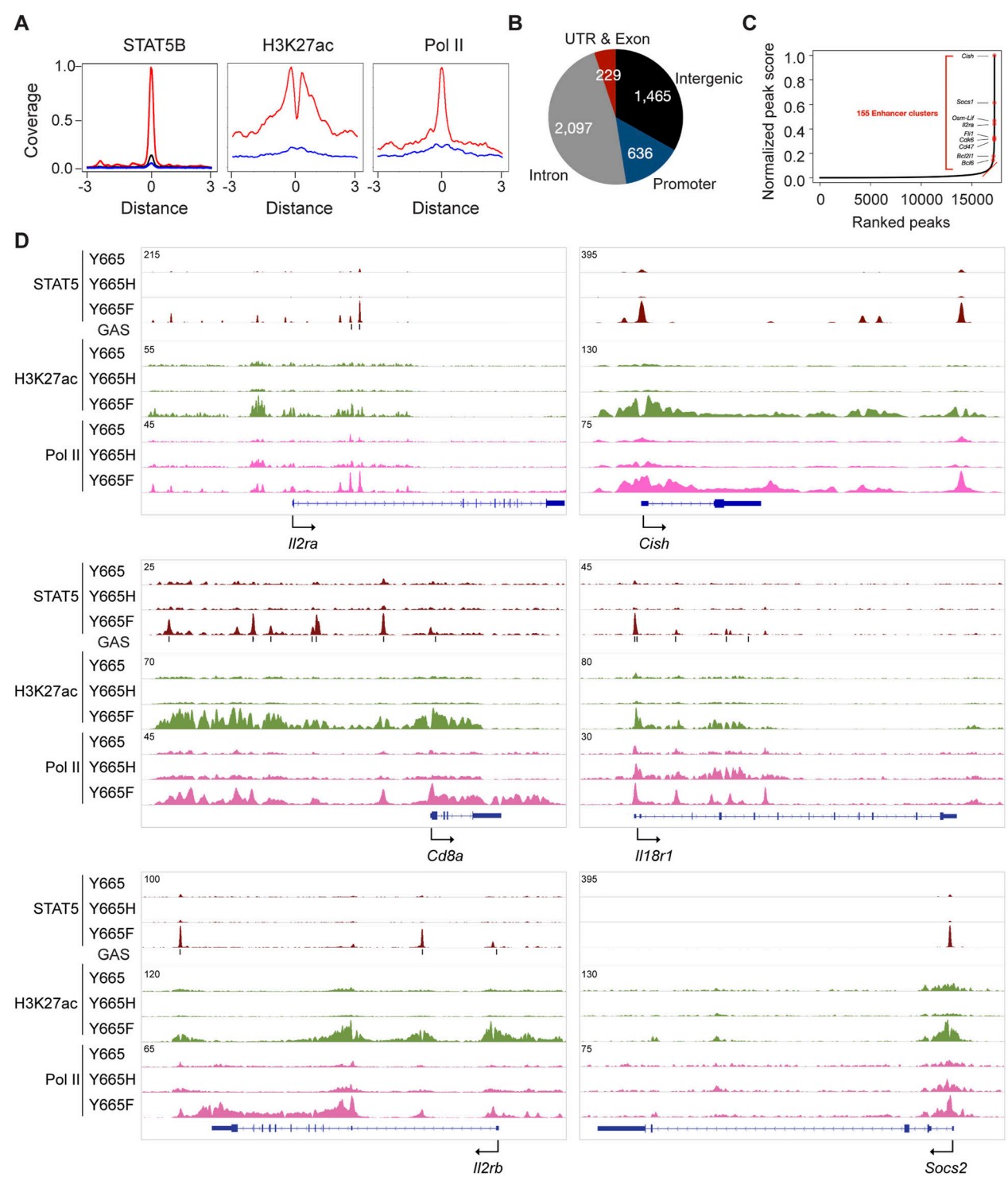

**Figure 6. Inverse impact of STAT5B^Y665F and STAT5B^Y665H on the establishment of transcription enhancers and immune programs.**
**(A)** Coverage plots displaying the patterns of STAT5B binding, H3K27ac marks, and Pol II loading on promoters of genes with or without STAT5B binding (blue, Y665H; red, Y665F; n = 2–3). **(B)** Distribution of STAT5B binding peaks across genomic regions. ChIP-seq signals within ± 3kbp of STAT5B binding regions identified in IL-2/IL-7–stimulated T cells of Y665F mice shown as pie chart. **(C)** Enhancer cluster analysis using IL-2/7–stimulated T cells of Y665F mice. The y-axis indicates the normalized peak score (enhancer cluster score) of STAT5B binding, whereas the x-axis represents the ranking of peaks. No enhancer clusters were identified in Y665H mice. **(D)** STAT5B binding, H3K27ac, and Pol II loading at regulatory elements of the interleukin-2 receptor subunit alpha (*Il2ra*), cytokine-inducible SH2–containing protein (*Cish*), cd8 subunit alpha (*Cd8a*), interleukin-18 receptor 1 (*Il18r1*), interleukin-2 receptor subunit beta (*Il2rb*), and suppressor of cytokine signaling 2 (*Socs2*) genes in IL-2/7–stimulated T cells from Y665, Y665H, and Y665F mice.

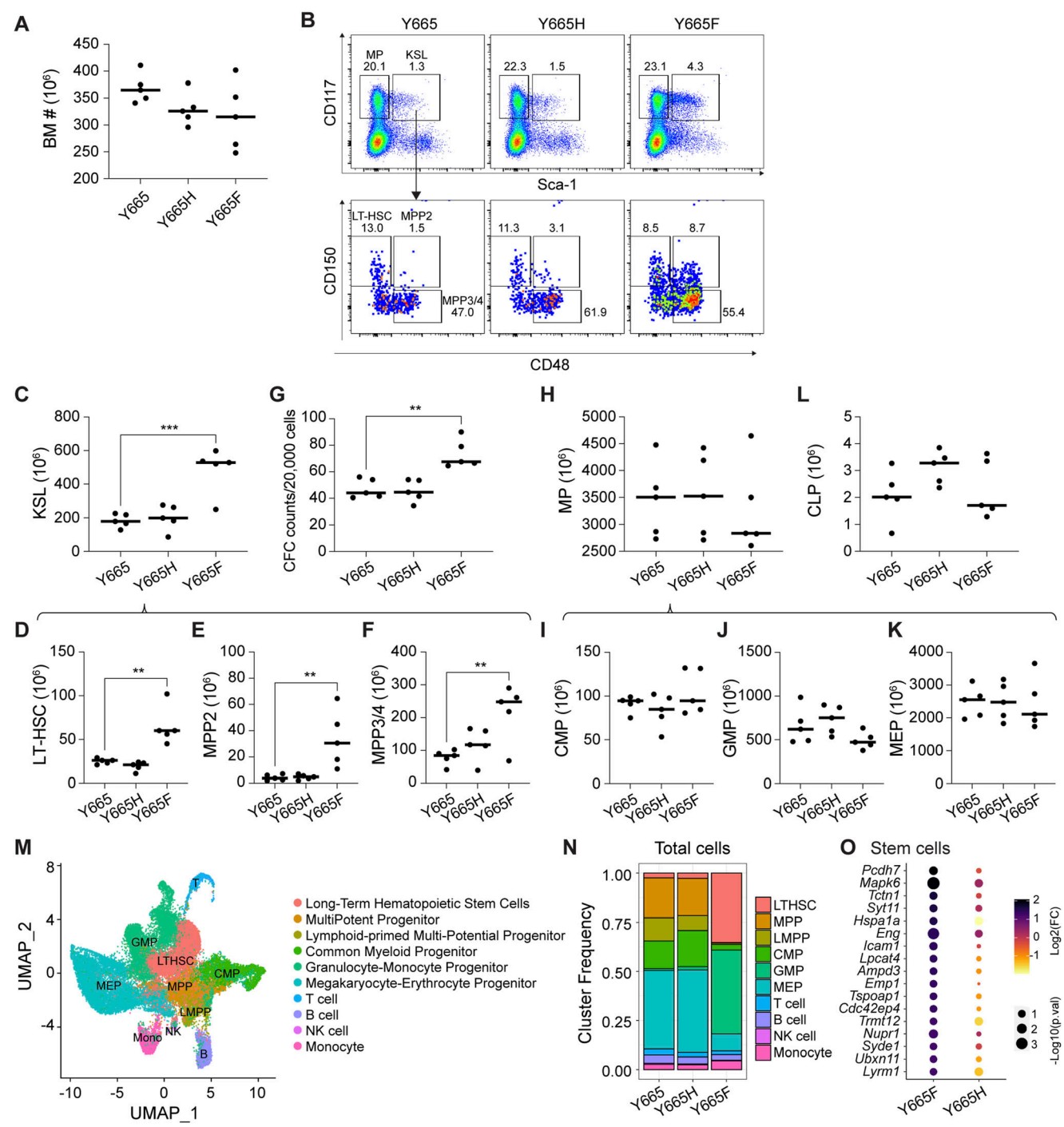

**Figure 7. Hematopoietic stem and progenitor cells in the bone marrow of Stat5b mutant mice.**
**(A)** Dot plot graph of total numbers of BM cells in *Stat5b*[Y665], *Stat5b*[Y665F], and *Stat5b*[Y665H] mice. **(B)** Representative flow cytometry plots of myeloid progenitors (Lin-CD117$^+$Sca-1$^-$), Lin-Sca-1$^+$CD117$^+$ cells (KSL), and KSL subpopulations. **(C)** Dot plot graph of absolute numbers of KSL (Lin-Sca-1$^+$CD117$^+$) cells. **(D)** Dot plot graph of absolute numbers of long-term hematopoietic stem (CD150$^+$CD48$^-$ KSL) cells. **(E)** Dot plot graph of absolute numbers of multipotent progenitor 2 (CD150$^+$CD48$^+$ KSL) cells. **(F)** Dot plot graph of absolute numbers of multipotent progenitor 3/4 (CD150$^-$CD48$^+$ KSL) cells. **(G)** Dot plot graph of colony-forming cell numbers per 20,000 BM cells. **(H)** Dot plot graph of absolute numbers of myeloid progenitor cells. **(I)** Dot plot graph of absolute numbers of common myeloid progenitor cells. **(J)** Dot plot graph of absolute numbers of granulocyte–monocyte progenitor cells. **(K)** Dot plot graph of absolute numbers of megakaryocytic erythroid progenitor cells. **(L)** Dot plot graph of absolute numbers of common lymphoid progenitor cells. **(M)** Uniform Manifold Approximation and Projection graph of the cell types identified from all three genotypes using single-cell sequencing. **(N)** Bar graphs of the cluster frequency of the different cell types identified in the bone marrow of *Stat5b*[Y665], *Stat5b*[Y665F], and *Stat5b*[Y665H] mice using single-cell sequencing. **(O)** Categorical scatter plot of relative gene expression values of representative genes from Gene Ontology (GO) category hematopoiesis stem cells in *Stat5b*[Y665F] and *Stat5b*[Y665H] mice. **(C, D, E, F, G, H, I, J, K, L)** Median of independent biological replicates shown as a horizontal line in panels (C, D, E, F, G, H, I, J, K, L) (*Stat5b*[Y665], n = 5; *Stat5b*[Y665F], n = 5; *Stat5b*[Y665H], n = 5). A one-way ANOVA followed by Tukey's multiple comparisons test was used to evaluate the statistical significance of differences between groups. *P < 0.05, **P < 0.01, ***P < 0.0001, ****P < 0.0001.

tail and the SH2 domain provided by substitution of a phenylalanine at position 665. Conversely, the reduced transcriptional activity of the STAT5B[Y665H] mutant is driven by the negative energetic contribution provided by substitution of a histidine at position 665. Our ability to identify high-confident STAT5B binding peaks in the ChIP-seq results from the cells of STAT5B[Y665F] mice, as compared to STAT5B[Y665] and STAT5B[Y665H] mice, is compatible with enhanced stabilization of the STAT5B[Y665F] dimer.

Even though both mutations studied have been identified as hemizygous mutants in patients with T-PLL (Rajala et al, 2013; Kiel et al, 2014; Kontro et al, 2014; Bhattacharya et al, 2022) and have confirmed GOF activity in vitro (Pham et al, 2018), there was no evidence of overt leukemia in homozygous *Stat5b*[Y665F] mice at 2 mo of age, as well as heterozygous *Stat5b*[Y665F] and homozygous *Stat5b*[Y665H] mice by 11 mo of age. However, some pathological features of STAT5B[Y665F] mice, such as an increase in CD8[+] and central memory cells, mirror that observed in T-LGLL patients. Notably, the hyperactive STAT5B[Y665F] galvanizes similar genetic programs in T-LGLL patients and in the mutant mice. The elevated expression of cytotoxicity-associated transcripts *Klrd1*, *Klrg1*, and *Gzma* is present in both patients (Rajala et al, 2013; Huuhtanen et al, 2022) and in mutant mice. We found that *Nkg7*, encoding the Natural Killer Cell Granule Protein 7, is a direct Stat5b target gene and highly induced in immune cells from *Stat5b*[Y665F] mice. NKG7 is expressed in CD4[+] and CD8[+] T cells and has shown to be important in the inflammatory process (Ng et al, 2020; Lelliott et al, 2022). Although STAT5B[Y665F] exerts transcriptional hyperactivity and was associated with HSC expansion in the mutant mice, no leukemia was observed within the timeframes examined. This does not exclude that leukemia might not develop if homozygous *Stat5b*[Y665F] mice could be aged out to longer timepoints, or heterozygous *Stat5b*[Y665F] and homozygous *Stat5b*[Y665H] mice to mouse ages beyond 12 mo. It is also possible leukemia could be revealed in bone marrow transplant experiments, which potentially could be engineered so that hematopoietic cell-autonomous effects would dominate without development of severe dermatitis seen in the homozygous *Stat5b*[Y665F] mice. BM transplant experiments would also be able to address questions about cell-intrinsic versus cell-extrinsic effects of the mutant STAT5B alleles.

In light of human data (Kiel et al, 2014) and in vitro studies (Pham et al, 2018), it was surprising that STAT5B[Y665H] was a LOF mutation and failed to activate genetic programs, in our both primary cell and mouse studies. AlphaFold3 predicted that the histidine substitution introduces a positively charged imidazole, potentially destabilizing the dimer is in full agreement with our findings. However, neither AlphaMissense, CADD, REVEL, nor PolyPhen-2 predicted a LOF, highlighting the need for experimental validation to accurately determine the pathophysiological effects in an in vivo setting. Congenital inactivating human STAT5B mutations, associated with severe growth failure and immune deficiencies, have been identified in several Laron syndrome patients (Kofoed et al, 2003; Hwa, 2021). Mutations such as STAT5B[A630P] (Kofoed et al, 2003) and STAT5B[F646S] (Scaglia et al, 2012) are autosomal recessive variants located within the SH2 domain, likely disrupting phosphate group binding on activated receptors or impairing STAT5 dimerization. Similar to STAT5B[Y665H], these mutations fail to activate transcription and are associated with T-cell lymphopenia, a phenotype also observed in our mutant mouse models.

This study emphasizes the importance of adopting a multidisciplinary approach to investigate the effects of mutations associated with human diseases. Although in silico tools like AlphaFold3 and COORDinator can predict structural impacts of mutations, functional algorithms such as AlphaMissense and PolyPhen-2, designed to assess pathogenicity, often produce inconsistent results. Similarly, in vitro cell culture studies provide only a limited perspective on the physiological and molecular consequences of mutations. A more comprehensive understanding of mutation pathogenicity can be achieved by integrating data from human patients with insights gained from experimental mouse genetics.

# Materials and Methods

### Mice

All animals were housed and handled according to the Guide for the Care and Use of Laboratory Animals (eighth edition), and all animal experiments were approved by the Animal Care and Use Committee (ACUC) of the National Institute of Diabetes and Digestive and Kidney Diseases (NIDDK, MD) and performed under the NIDDK animal protocol K089-LGP-20. CRISPR/Cas9 and base editing–targeted mice were generated using C57BL/6N mice (Charles River) by the Transgenic Core of the National Heart, Lung, and Blood Institute (NHLBI). Single-guide RNAs (sgRNA) were obtained using Thermo Fisher Scientific's In Vitro Transcription Service (Table S6). A single-strand oligonucleotide donor was obtained from IDT (Table S6). For the *Stat5b*[Y665H] mutant mice, the ABE mRNA (50 ng/$\mu$l) and Y665H sgRNA (20 ng/$\mu$l) were co-microinjected into the cytoplasm of fertilized eggs collected from superovulated C57BL/6N female mice (Charles River Laboratories). For the *Stat5b*[Y665F] mutant mice, a single-strand oligonucleotide donor contained the desired Y (TAC)-to-F (TTT) change and a silent C-to-G change. The silent mutation does not result in amino acid change but can destroy the sgRNA PAM and hence stopping Cas9 from further cutting after the oligo template was successfully knocked in. The *Stat5b*[Y665F] sgRNA was first mixed with Cas9 protein (IDT) to form Cas9 RNP complex, which was co-electroporated with the oligo template into zygotes collected from C57BL/6N mice using a Nepa21 electroporator (Nepa Gene Co) following procedures described by Kaneko (2023). The microinjected or electroporated zygotes were cultured overnight in M16 medium (MilliporeSigma) at 37°C with 6% $CO_2$. Those embryos that reached two-cell stage of development were implanted into the oviducts of pseudopregnant surrogate mothers (Swiss Webster mice from Charles River). All mice born to the foster mothers were genotyped by PCR amplification and Sanger sequencing (Quintara Biosciences) and automated genotyping using a TaqMan-based assay (TransnetYX) with genomic DNA from mouse tails. Mice aged 2–11 mo were used in experiments. Mice aged 2 mo were used for experiments examining immune cell function and early hematopoiesis in the BM. Tissues were collected from 2-mo-old male mice and used immediately or stored at –80°C. Dermatitis was monitored throughout the life of the mice, and

weights were measured until age 10 wk. For experiments examining the peripheral blood, mice aged 2 (Stat5b[Y665], heterozygous Stat5b[Y665F], homozygous Stat5b[Y665F], homozygous Stat5b[Y665H]) to 11 (Stat5b[Y665], heterozygous Stat5b[Y665F], homozygous Stat5b[Y665H]) mo were used. Age of death was defined as the age when mice were found dead in a cage or underwent humane euthanasia for life-threatening dermatitis.

## Whole-exome sequencing and data analysis

Genomic DNA was isolated from tail tissue using Wizard Genomic DNA Purification Kit (Promega). Exome sequencing and bio-informatics analyses were performed at Psomagen. Target capture for the exome was performed on each sample using SureSelect Mouse All Exon Kit (Agilent Technologies). DNA was subjected to SureSelect Target Enrichment System for paired-end DNA library preparation. Whole-exome sequencing was performed on a NovaSeq 6000 instrument (Illumina).

Sequencing reads were aligned to the mouse reference (mm10) using BWA 0.7.10. After excluding chimeric reads, the duplicated reads were eliminated using Picard. GATK3.v4 "IndelRealigner" and "Table Recalibration" were used for local realignment and for recalibrating the quality scores, respectively. For SNV/Indel calling in multisample analysis, GATK "HaplotypeCaller" was used for comparison with the reference genome. For SNV calling in matched-pair analysis, "Selectvariants" was used to compare the difference between WT and mutant mice. Annotation for all variants was made using dbSNP142.

## Cell counts and flow cytometry

Blood was collected from retro-orbital sinus into Eppendorf tubes in the presence of 5 mM EDTA (Sigma-Aldrich). Complete blood counts were performed using an Element HT5 analyzer (Heska Corporation).

Mice were euthanized by $CO_2$ to collect spleen, lymph node, tibia, and femur. Spleens and lymph nodes were mechanically dissociated and filtered through a 70-micron strainer. BM cells were extracted from bilateral tibiae and femurs, filtered through 85-micron nylon mesh, and number-counted by Vi-Cell counter (Beckman Coulter). Total BM cells per mouse were calculated based on the assumption that 2 tibiae and 2 femurs contain 25% of total BM cells. To stain cell surface antigens, peripheral blood, spleen, and BM cells were first incubated in ACK buffer to remove red blood cells and were then stained with antibody mixtures for flow cytometry analysis.

After cell surface and fixable viability staining of single-cell suspensions with fluorophore-conjugated antibodies in PBS, in-tracellular staining was performed after fixing and permeabilizing cells with the Foxp3/TF Staining kit (eBioscience) per manufac-turer's instructions. For measuring levels of phosphorylated STATs after in vitro stimulation, single-cell suspensions were stained with cell surface fluorophore-conjugated antibodies and fixable viability dye in PBS, fixed for 20 min at 4°C with BD Fix/Perm solution (Cat #554722), permeabilized with 100% cold methanol for 20 min at 4°C, and stained with anti-p-STAT5 fluorophore-conjugated antibodies in 1xPerm/Wash Buffer (#557885; BD Biosciences) for 30 min at room temperature. The following antibodies were used for flow cytom-etry: BUV395-CD19 (ID3), BUV496-CD4 (GK1.5), BUV737-CD62L (MEL-14), BUV805-gdTCR (GL3), BV421-CD8a (53-6.7), eFluor450-Foxp3 (FJK-16s), BV480-TCRb (H57-597), BV510-CD11b (M1/70), BV570-CD44 (IM7), BV605-CD25 (PC61), AF532-CD45 (30-F11), Spark Blue 550-Ly6G (1A8), PerCP-eFluor710-NKp46 (29A1.4), AF647-pSTAT5 (C71E5), Zombie NIR, APC-NK1.1 (PK136), CD3, CD4, CD8, CD45R, Gr-1, CD11b, Ter119, CD117, Sca-1, CD48, CD150, CD34, CD16/32 (BioLegend). 7AAD and annexin V were purchased from BD Biosciences. Stained cells were acquired using BD FACSCanto II and BD LSRFortessa flow cytometry operated by FACSDiva software (Becton Dickson) and 5-laser Cytek Aurora performed with FlowJo v10.

## Colony-forming assay

BM cells from Stat5b[Y665], Stat5b[Y665F], and Stat5b[Y665H] mice were mixed in semisolid methylcellulose medium (MethoCult GF M3434, Cat# 03444; STEMCELL Technologies) containing interleukin (IL)-3, IL-6, stem cell factor, and erythropoietin (EPO), and plated on 35-mm culture dishes at $2 \times 10^4$/plate. Cells were cultured at 37°C with 5% $CO_2$. Colonies were counted at day 10.

## T-cell isolation and in vitro stimulation

For the pSTAT5 level, lymphocytes isolated from spleen were stimulated at 37°C in complete RPMI alone or with 1,000 U/ml of recombinant mouse IL-2 (R&D Systems), 10 ng/ml of IL-7 (R&D Systems), and/or 20 ng of IL-15 (R&D Systems) for 30 min.

For RNA-seq and ChIP-seq, total T cells from isolated splenocytes were isolated using EasySep Mouse T-cell Isolation Kit (#19851; STEMCELL) and cultured with recombinant mouse IL-2 and IL-7 for 24 h.

## Isolation of Stat5a/b-deficient CD4 T cells

The Cd4-Cre transgene was introduced into Stat5a/b floxed mice (Cui et al, 2004) resulting in the deletion of the Stat5 locus in both CD4 and CD8 cells at the "double-positive" stage of thymic de-velopment. The starting population are sorted naïve CD4[+] T cells (live, CD4[+]CD44 lowCD25[−]) from poled lymph nodes and spleens. The sequenced population are sorted RV-transduced CD4[+] T cells (live, CD4[+]GFP[+]). Cells were transduced with an empty retroviral vector or vectors encoding the native and mutant STAT5B isoforms cultured for 48 h with IL-2 and subsequently subjected to FACS analysis and RNA-seq. T cells are activated with anti-TCR and anti-CD28 anti-bodies before retroviral transduction. The isoform of STAT5 that was packaged into retrovirus was Stat5b-202 (ENSMUST00000107358.9), classified by Ensembl as "canonical". This was then used as a template for the site-directed mutagenesis to produce the Stat5b variants.

## Total RNA-seq and data analysis

Total RNA was extracted from either frozen spleen tissue or total T cells isolated from total splenocytes from WT and mutant mice and purified with RNeasy Plus Mini Kit (74134; QIAGEN). Ribosomal RNA was removed from 1 µg of total RNAs, and cDNA was

synthesized using SuperScript III (Invitrogen). Libraries for sequencing were prepared according to the manufacturer's instructions with TruSeq Stranded Total RNA Library Prep Kit with Ribo-Zero Gold (RS-122-2301; Illumina), and paired-end sequencing was done with a NovaSeq 6000 instrument (Illumina).

Total RNA-seq read quality control was done using Trimmomatic (Bolger et al, 2014) (version 0.36), and STAR RNA-seq (Dobin et al, 2013) (version STAR 2.5.4a) using paired-end mode was used to align the reads (mm10). HTSeq (Anders et al, 2015; Li et al, 2020) was used to retrieve the raw counts, and subsequently, R (https://www.r-project.org/), Bioconductor (Huber et al, 2015), and DESeq2 (Love et al, 2014) were used. In addition, the RUVSeq (Risso et al, 2014) package was applied to remove confounding factors. The data were pre-filtered keeping only those genes, which have at least ten reads in total. Genes were categorized as significantly differentially expressed with log2 fold change >1 or < −1, and adjusted P-values (Padj) < 0.05 corrected for multiple testing using the Benjamini–Hochberg method were considered significant, and then, we conducted gene enrichment analysis (Metascape; https://metascape.org/gp/index.html#/main/step1). The visualization was done using dplyr (https://CRAN.R-project.org/package=dplyr) and ggplot2 (Wickham, 2009).

### scRNA-seq and data analysis

Single-cell suspensions were then immediately loaded on the 10X Genomics Chromium Controller with a loading target of 10,000 cells. Libraries were generated using Chromium Next GEM Single Cell 5′ Kit v2 (Dual Index) and Chromium Single Cell Mouse TCR Kit 5′ v2 (Dual Index) according to the manufacturer's instructions. Libraries were sequenced using the NovaSeq 6000 instrument (Illumina).

The raw reads were aligned and quantified using the Cell Ranger with Feature Barcode addition (version 6.0, 10X Genomics) against the GRCm38 mouse reference genome. The quality control, normalization, dimension reduction, cell clusters, Uniform Manifold Approximation and Projection, and cell type annotation were performed using Seurat (version 4.3) (Hao et al, 2021).

### Chromatin immunoprecipitation sequencing (ChIP-seq) and data analysis

The frozen–stored tissues were ground into powder in liquid nitrogen. Chromatin was fixed with formaldehyde (1% final concentration) for 15 min at room temperature and then quenched with glycine (0.125 M final concentration). Samples were processed as previously described (Metser et al, 2016). The following antibodies were used for ChIP-seq: STAT5B (AF1584; R&D Systems; 13-5300; Thermo Fisher Scientific), H3K27ac (ab4729; Abcam), and RNA polymerase II (ab5408; Abcam). Libraries for next-generation sequencing were prepared and sequenced with the NovaSeq 6000 instrument (Illumina).

Quality filtering and alignment of the raw reads were done using Trimmomatic (Bolger et al, 2014) (version 0.36) and Bowtie (Langmead et al, 2009) (version 1.2.2), with the parameter "-m 1" to keep only uniquely mapped reads, using the reference genome mm10. Picard tools (Broad Institute. Picard, http://broadinstitute.github.io/picard/. 2016) were used to remove duplicates, and subsequently,

Homer (Heinz et al, 2010) (version 4.9.1) software and deepTools (Ramirez et al, 2016) (version 3.1.3) software were applied to generate bedGraph files and normalize coverage, separately. Integrative Genomics Viewer (Thorvaldsdottir et al, 2013) (version 2.3.98) was used for visualization. Coverage plots were generated using Homer (Heinz et al, 2010) software with the bedGraph from deepTools (Ramirez et al, 2014) as output. R and the packages dplyr (https://CRAN.R-project.org/package=dplyr) and ggplot2 (Love et al, 2014) were used for visualization. Each ChIP-seq experiment was conducted for two replicates, and the correlation between the replicates was computed by the Spearman correlation using deepTools.

To identify high-confident STAT5B binding peaks, we defined the STAT5B binding peaks that coincide with H3K27ac marks within ±500 bp and contain at least one GAS motif (5′-TTCNNNGAA-3′) within the peak region. The Integrative Genomics Viewer (IGV) tools were used for ChIP-seq signal visualization. Tag density plots and heatmaps were generated using computeMatrix and plotHeatmap tools implemented in deepTools (version 3.1.3). To examine enhancer clusters, we used findPeaks.pl with the following parameter settings: -style super -superSlope −1,000. Enhancer clusters were identified as regions with "slope" (focus ratio)/(region size annotation enhancer) greater than 1. For further analyses, only protein-coding genes were used.

### Protein structure prediction and analysis

The STAT5B dimer structure was predicted using AlphaFold3 (https://alphafoldserver.com). The input sequences included human STAT5B (UniProt accession P51692) and the DNA fragment sequence (5′-GTTTCTTCTGAGAAGTACC-3′) derived from STAT5-bound Il2ra enhancer. The structure prediction was performed with pY699 phosphorylation parameter. Structural analysis and visualization were performed using PyMOL (version 2.4.1) and ChimeraX (version 1.8).

The dimeric structures of human STAT5A (UniProt accession P42229, residues 620–715) and STAT5B (UniProt accession P51692, residues 620–720) with phosphorylated residues STAT5A$^{pY694}$ and STAT5B$^{pY699}$ were modeled using AlphaFold3 (Abramson et al, 2024). The energetic consequences of amino acid substitutions in STAT5A and STAT5B were predicted by COORDinator using these structures (Li et al, 2023). COORDinator mutation predictions were made by calculating the difference between the COORDinator-predicted energies (in arbitrary units) for the mutant and WT residues. COORDinator was fine-tuned to enforce a high correlation between predicted and experimental protein stability measurements for hundreds of thousands of mutations (Tsuboyama et al, 2023). The impact of a mutation on dimerization was predicted using the difference between the COORDinator energies for a residue in the absence versus the presence of the dimerization partner, without modeling any change in conformation. Predicted effects of mutations on intramolecular interactions with the C-terminal tail were obtained similarly, by treating the C-terminal tail as a separate chain. For example, the COORDinator-predicted energies for residues in the STAT5B C-terminal tail (residues 693–720) were computed in the context of the tail alone or in the context of the STAT5B monomer, using the conformations predicted for the STAT5B dimer.

Similarly, energies for residues in the SH2 domain (residues 620–687) were computed in the presence versus the absence of the C-terminal tail. When treating the STAT5B C-terminal tail as a separate chain, residues 688–692 were not included, to avoid effects from the amino- and carboxyl-groups of the artificially separated chains. The predicted importance of a residue was derived by summing the absolute values of the COORDinator energy predictions for all mutations at that site. For this analysis, we used code from COORDinator GitHub (https://github.com/daveneff/Coordinator) and Terminator GitHub (https://github.com/KeatingLab/terminator).

### In silico pathogenicity prediction

Multiple computational tools were employed to assess the potential pathogenicity of STAT5B variants. AlphaMissense (Cheng et al, 2023) was used to predict the functional impact of mutations, with scores ranging from 0 (benign) to 1 (pathogenic). The CADD (Rentzsch et al, 2019) tool was used to estimate the deleteriousness of variants, where PHRED scores above 20 indicate potentially damaging effects. The REVEL (Ioannidis et al, 2016) scores, which range from 0 to 1, were calculated to assess the probability of pathogenicity. In addition, PolyPhen-2 (Adzhubei et al, 2013) analysis was performed to predict the possible impact of amino acid substitutions on protein structure and function, with scores ranging from 0 (benign) to 1 (probably damaging).

### Statistical analyses

All samples that were used for complete blood counts, FACS, and RNA-seq were randomly selected, and blinding was not applied. For comparison of samples, data were presented as SD in each group and were evaluated with one- or two-way ANOVA multiple comparisons using PRISM GraphPad. Statistical significance was obtained by comparing the measures from the WT or the control group, and each mutant group. A value of $*P < 0.05$, $**P < 0.001$, $***P < 0.0001$, $****P < 0.00001$ was considered statistically significant; ns, no significant.

## Data Availability

All data were obtained or uploaded to Gene Expression Omnibus (GEO). ChIP-seq, RNA-seq, and scRNA-seq data of WT and mutant tissues are under GSE276308 (secure token: wlcremgmtvkfxuz), GSE276311 (secure token: mlkxeassfvkbzcf), and GSE276312 (secure token: clqlowasdvydpin). The fastq files and the processed *bedGraph* files can be downloaded from GEO (https://www.ncbi.nlm.nih.gov/gds/?term=) and imported into the IGV browser (https://software.broadinstitute.org/software/igv/download) with a reference genome (mm10).

## Supplementary Information

## Acknowledgements

We thank the NHLBI sequencing core for NGS. This work used the computational resources of the NIH HPC Biowulf cluster (http://hpc.nih.gov). HK Lee, S-G Lee, PA Furth, and L Hennighausen were supported by the Intramural Research Programs (IRPs) of the National Institute of Diabetes and Digestive and Kidney Diseases (NIDDK); J Chen, X Feng, Z Wu, C Liu, and NS Young were supported by Intramural Research Programs (IRPs) of National Heart, Lung, and Blood Institute (NHLBI); and RL Philips and JJ O'Shea were supported by Intramural Research Programs (IRPs) of National Institute of Arthritis and Musculoskeletal and Skin Diseases (NIAMS). AB Schultz was supported by NIH Training grant T32AI162624. JA Sexton and F Birnbaum were supported by the National Institute of General Medical Sciences of the NIH under R35 GM149227 to AE Keating. AV Villarino was supported by the V Foundation (grant DEC2024-010), The Leukemia Research Foundation (grant AWD 009890) and the Universtiy of Miami, Department of Immunology and Microbiology grant (PG013596) and the Sylvester Comprehensive Center start up grant (PG012707).

## Author Contributions

HK Lee: conceptualization, resources, data curation, formal analysis, investigation, visualization, methodology, project administration, and writing—original draft, review, and editing.
J Chen: data curation, formal analysis, validation, investigation, visualization, and methodology.
RL Philips: formal analysis, validation, investigation, visualization, and methodology.
S-G Lee: formal analysis.
X Feng: formal analysis, investigation, and methodology.
Z Wu: investigation and methodology.
C Liu: conceptualization, resources, and funding acquisition.
AB Schultz: formal analysis and investigation.
M Dalzell: formal analysis and investigation.
M Meggendorfer: data curation and investigation.
C Haferlach: data curation, investigation, and project administration.
F Birnbaum: formal analysis.
JA Sexton: formal analysis.
AE Keating: conceptualization and formal analysis.
JJ O'Shea: conceptualization and funding acquisition.
NS Young: conceptualization and funding acquisition.
AV Villarino: conceptualization, formal analysis, funding acquisition, investigation, visualization, methodology, and writing—original draft, review, and editing.
PA Furth: conceptualization, data curation, supervision, investigation, project administration, and writing—original draft, review, and editing.
L Hennighausen: conceptualization, resources, data curation, supervision, funding acquisition, investigation, project administration, and writing—original draft, review, and editing.

### Conflict of Interest Statement

The authors declare that they have no conflict of interest.

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
