## [Reviewer comments · Life Science Alliance]

Life Science Alliance

STAT5B leukemic mutations, altering SH2 tyrosine 665, have opposing impacts on immune gene programs

Hye Kyung Lee, Jichun Chen, Rachael Philips, Sung-Gwon Lee, Xingmin Feng, Zhijie Wu, Chengyu Liu, Aaron Schultz, Molly Dalzell, Manja Meggendorfer, Claudia Haferlach, Foster Birnbaum, Joel Sexton, Amy Keating, John O'Shea, Neal Young, Alejandro Villarino, Priscilla Furth, and Lothar Hennighausen

DOI: <https://doi.org/10.26508/lsa.202503222>

Corresponding author(s): *Lothar Hennighausen, National Institute of Diabetes and Digestive and Kidney Diseases*

Review Timeline:

Submission Date:	2025-01-14
Editorial Decision:	2025-03-14
Revision Received:	2025-03-23
Editorial Decision:	2025-03-24
Revision Received:	2025-03-31
Accepted:	2025-03-31

Transaction Report:

March 14, 2025

Re: Life Science Alliance manuscript #LSA-2025-03222-T

Dr. Lothar Hennighausen
National Institute of Diabetes, Digestive and Kidney Diseases
8 Center Dr
Bethesda, MD 20892

Dear Dr. Hennighausen,

Thank you for submitting your manuscript entitled "STAT5B leukemic mutations, altering SH2 tyrosine 665, have opposing impacts on immune gene programs" to Life Science Alliance. The manuscript was assessed by expert reviewers, whose comments are appended to this letter. We invite you to submit a revised manuscript addressing the Reviewer comments.

Thank you for this interesting contribution to Life Science Alliance. We are looking forward to receiving your revised manuscript.

Sincerely,

B. MANUSCRIPT ORGANIZATION AND FORMATTING:

Reviewer #1 (Comments to the Authors (Required)):

In the manuscript by Hennighausen et al (LSA-2025-03222-T), authors addressed effects of two amino-acid replacements at Y665 residue in STAT5B, Y665F and Y665H, both were isolated from human T cell leukemia. Starting from structural impact using in silico simulation, authors tested impact of retroviral transduction of these two forms in Stat5a/b^{-/-} CD4⁺ cells. Furthermore, authors generated mouse models by introducing these amino acid change by genome editing. Results using these materials and approaches showed that STAT5B-Y665F behaves like a gain-of-function form, while STAT5B-Y665H showed loss-of-function form.

Overall, this work is designed well and results shown supported authors conclusion. Thus, I support to publication of this manuscript for in Life Science Alliance after minor revision.

Major comments:

1. It is better to show representative histogram for pATAT5 staining in Figure 5A-B.
2. Please add paragraph that discusses how Y665F replacement led to a gain-of-function and possibility for F residue may mimics phosphorylation of Tyrosine in this case.

Reviewer #2 (Comments to the Authors (Required)):

The paper by Lee et al investigates the impact of two STAT-5 mutations, Y665F, detected in several cases of T-cell and Y665H, found only in a single case of leukemia. The authors investigate the functional activity of these mutants by in silico analysis, overexpression of the proteins on a STAT5a/b deficient background and by germline KI in mice. The in vitro predictions of pathogenicity were somewhat different dependent on the tools used but generally the F mutation, predicted from structure analysis to cause stabilization of dimerization, received a higher pathogenicity score. Over expression of the mutant proteins on a Stat5a/b deficient background supported the idea that the F mutation was activating the H mutation inhibited protein activity. Introduction of these mutations in the mouse germline resulted in alterations in T-cell populations as well as, in the case of the Y665F mutation, splenomegaly and dermatitis. The authors use single cell RNA seq as well as Chip-seq to generate further support for the notion that Y665F represents a gain and Y665H results in loss of function.

The manuscript is well written, and the authors use a large number of techniques to provide support the general conclusion with regard to gain and loss of function of the mutations. My major concern comes from the use of germline mutations making it difficult to determine if the observed effects represents a cell autonomous developmental defect or a consequence of an ongoing inflammation in the Y665F mice. While this does not impact the conclusions drawn regarding the general functionality of these mutations, it complicates the interpretations of the experiments exploring the biological impact of these mutations.

Specific comments:

1. The impact of this paper would be dramatically increased if the authors performed transplantation experiments transferring mutated BM cells (LSK cells to avoid transfer of activated T-cells) into congenic Wt mice. This would allow for much stronger in vivo data and might allow the authors to explore leukemia development as these animals might not develop dermatitis.
2. Previous reports (Pham et al 2017) indicate that over expression of the Y665F mutations disturbs early hematopoiesis. It would be most interesting to see how early hematopoiesis is affected by these mutations. Even though this could be done in the germline mutants, the data would be stronger if collected from a transplantation model.
3. Figure 3B indicates survival rates, perhaps not correct term when the animals are being euthanized.
4. In the discussion row 355, the authors claim that there were no signs of leukemia even at 11 months. Perhaps not supported by the data as all the Y665F mice were euthanized at 6 months (24 weeks).

Reviewer #3 (Comments to the Authors (Required)):

Lee et al. demonstrate in this paper that two different STAT5B mutations show opposing function by in vivo and in vitro experiment. Authors showed STAT5B-Y665F is a gain of function mutation in contrast to STAT5B-Y665H which shows loss of function. Authors examined the immune-phenotype of STAT5B mutation mouse and provided data to support the hypothesis that STAT5B-Y665F mutation could increase the STAT5B phosphorylation, transcriptional activities. The manuscript is very interesting, but several important concerns need to be addressed.

Major concerns:

- 1- Authors discovered two STAT5B mutations from human T cells leukemia and analyzed the predicated structural in silico. To test these two mutations' function, two mutation mouse models were introduced. Authors did not clarify if STAT5B-Y665 amino acid is conserved or not across different species. And which kind of STAT5B was transfected into mouse CD4T cells was not clear.
- 2- In Fig2, authors transduced different mutants of STAT5B retrovirus vector into WT or Stat5a/b-DKO mice to test the function of STAT5B. According to the data showed in Fig2C, WT-STAT5B could not fully rescue the Stat5a/b DKO phenotype as showed by the CD25 expression level. Please explain the reason.
- 3- In Fig3, authors showed increase activated CD8T cells and increase percentage of Treg cells in STAT5B-Y655F mice. What's the mechanism for increased both effector and regulatory cell subsets? Is this effect due to a cell-intrinsic or cell-extrinsic manner?
- 4- IL2R(CD25) is the most important downstream gene regulated by STAT5B. It would be better to show the basic CD25 expression level in different mutant's mice model. And it is well known that naïve T cells are majorly CD25-negative. So it is meaningless to test pSTAT5 level after 30min in vitro stimulation, as CD25 is not expressed in naïve T cells.
- 5- Authors used total T cells for histone modification and TF binding analysis. Considering the altered cell subsets, the difference showed in Fig 6, the majoring reason of the difference could be explained by different contributing from different cells subset but not its binding activities.

Minor concerns:

- 1- It is not convincing to use Q-PCR result indicate the expression level of retrovirus expression level. Western blot would be better to indicate similar level of proteins were introduced in Stat5a/b-DKO cells
- 2- It would be better to show the gating strategy in the sFigures.
- 3- There are some typos, e.g. In Fig6D, "F665H" should be "Y655H". Please double check the manuscript

We appreciate all three reviewers' positive comments about the work:

Reviewer #1: "...Overall, this work is designed well and results shown supported authors conclusion. Thus, I support to publication of this manuscript for in Life Science Alliance after minor revision."

Reviewer #2: "... The manuscript is well written, and the authors use a large number of techniques to provide support the general conclusion with regard to gain and loss of function of the mutations."

Reviewer #3: "...The manuscript is very interesting,..."

Our detailed response to each reviewer's comments are addressed below.

Reviewer #1 (Comments to the Authors (Required)):

In the manuscript by Hennighausen et al (LSA-2025-03222-T), authors addressed effects of two amino-acid replacements at Y665 residue in STAT5B, Y665F and Y665H, both were isolated from human T cell leukemia. Starting from structural impact using in silico simulation, authors tested impact of retroviral transduction of these two forms in Stat5a/b-/- CD4+ cells. Furthermore, authors generated mouse models by introducing these amino acid change by genome editing. Results using these materials and approaches showed that STAT5B-Y665F behaves like a gain-of-function form, while STAT5B-Y665H showed loss-of-function form.

Overall, this work is designed well and results shown supported authors conclusion. Thus, I support to publication of this manuscript for in Life Science Alliance after minor revision.

Major comments:

1. It is better to show representative histogram for pSTAT5 staining in Figure 5A-B.

Response

Thank you. Representative histograms for pSTAT5 staining are presented in a new panel (Figure 5A).

2. Please add paragraph that discusses how Y665F replacement led to a gain-of-function and possibility for F residue may mimics phosphorylation of Tyrosine in this case.

Response

This is a good point, thank you. We have added a paragraph (Discussion section, lines 403-424) as well as adding lines 399-400 in the Discussion section that address "*how Y665F replacement led to a gain-of-function and possibility for F residue may mimics phosphorylation of Tyrosine in this case*". Text has been added to the Results section

(lines 328-331) to address this question, and a new reference (Weaver and Silva, 2006) added to clarify that while phosphorylation of tyrosine residues 699, 725, 740 and 743 are reported to influence STAT5b function, there are no reports that the same is true of STAT5B tyrosine 665 (lines 411-415).

Reviewer #2 (Comments to the Authors (Required)):

The paper by Lee et al investigates the impact of two STAT-5 mutations, Y665F, detected in several cases of T-cell and Y665H, found only in a single case of leukemia. The authors investigate the functional activity of these mutants by in silico analysis, overexpression of the proteins on a STAT5a/b deficient background and by germline KI in mice. The in vitro predictions of pathogenicity were somewhat different dependent on the tools used but generally the F mutation, predicted from structure analysis to cause stabilization of dimerization, received a higher pathogenicity score. Over expression of the mutant proteins on a Stat5a/b deficient background supported the idea that the F mutation was activating the H mutation inhibited protein activity. Introduction of these mutations in the mouse germline resulted in alterations in T-cell populations as well as, in the case of the Y665F mutation, splenomegaly and dermatitis. The authors use single cell RNA seq as well as Chip-seq to generate further support for the notion that Y665F represents a gain and Y665H results in loss of function.

The manuscript is well written, and the authors use a large number of techniques to provide support the general conclusion with regard to gain and loss of function of the mutations. My major concern comes from the use of germline mutations making it difficult to determine if the observed effects represent a cell autonomous developmental defect or a consequence of an ongoing inflammation in the Y665F mice. While this does not impact the conclusions drawn regarding the general functionality of these mutations, it complicates the interpretations of the experiments exploring the biological impact of these mutations.

Specific comments:

1. The impact of this paper would be dramatically increased if the authors performed transplantation experiments transferring mutated BM cells (LSK cells to avoid transfer of activated T-cells) into congenic Wt mice. This would allow for much stronger in vivo data and might allow the authors to explore leukemia development as these animals might not develop dermatitis.

Response

Thank you for the valuable suggestion. We agree that “*transplantation experiments transferring BM cells (LSK to avoid transfer of activated T-cells)*” would potentially allow for the exploration of leukemia development in homozygous STAT5B^{Y665F} at older ages. This is because transplanted mice might or might not develop the same dermatitis that

limited the study of leukemia development in the homozygous STAT5B^{Y665F} mice to younger ages. Regrettably, due to current funding limitations at NIH, bone marrow transplant experiments into congenic wild-type mice cannot be undertaken at this time. We have added text to discuss the value of bone marrow transplant experiments to the Discussion section (lines 443-446). We added lines 438-443 to the Discussion section to discuss the potential value of aging heterozygous STAT5B^{Y665F} and homozygous STAT5B^{Y665H} mice beyond the 11 months of age timepoint used in the present study to observe for development of leukemia.

2. Previous reports (Pham et al 2017) indicate that over expression of the Y665F mutations disturbs early hematopoiesis. It would be most interesting to see how early hematopoiesis is affected by these mutations. Even though this could be done in the germline mutants, the data would be stronger if collected from a transplantation model.

Response

Thank you. The report by Pham et al. was published in final form in the Journal of Clinical Investigation (JCI) in 2018. Relative activation levels of STAT5B^{N642H}, STAT5B^{Y665F} and STAT5B^{Y665H} in 293T kidney cells stimulated with growth hormone are presented: *"We observed high pY-STAT5 levels under steady-state conditions in cells expressing the N642H mutation, the most frequent STAT5 mutation in patients with leukemia or lymphoma. The 2 SH2 domain variants Y665H and Y665F also showed enhanced activity in the absence of cytokine stimulation (Figure 1B)."* Pham et al (2018) developed hSTAT5B^{N642H} transgenic mice (e.g. *Vav-1-hSTAT5B^{N642H}*) but not lines of mice expressing either STAT5B^{Y665F} or STAT5B^{Y665H}. The *Vav-1-hSTAT5B^{N642H}* mice developed malignant disease including *"leukemic cells"* within 40-100 days of life. Mice that were transplanted with bone marrow from the *Vav-1-hSTAT5B^{N642H}* mice showed: *"a phenotype comparable to that of hSTAT5B^{N642H}-transgenic mice. The disease was characterized by enlarged spleens and lymphoma formation, with T cell infiltration into peripheral organs (Figure 5B and Supplemental Figure 5, A and B) caused by excessive expansion and infiltration of CD8+ T cells (Figure 5, C and D)...The recipients of mutant cells became terminally sick approximately 3 months after injection."* Thus the time frame of disease development was similar for both the transgenic and bone marrow transplanted mice (e.g. ~90-days). Pham et al. (2018) examined early hematopoiesis in the bone marrow of the *Vav-1-hSTAT5B^{N642H}* transgenic mice. They found expansion of hematopoietic stem cells (HSC).

In the revised manuscript we added an experiment examining early hematopoiesis in the bone marrow of *Stat5b^{Y665}*, *Stat5b^{Y665F}*, and *Stat5b^{Y665H}* mice (Results section 350-376, new Figure 7, Discussion section (lines 438-440), Methods section lines 519-520, 551, 553, 555-556, 579-584). We found hematopoietic stem cell (HSC) expansion in the bone marrow of the *Stat5b^{Y665F}* mice. *Stat5b^{Y665H}* mice did not show HSC expansion.

3. Figure 3B indicates survival rates, perhaps not correct term when the animals are being euthanized.

Response

Thank you. Figure 3B has been re-labeled as “age of death” (Methods section, lines 528-529).

4. In the discussion row 355, the authors claim that there were no signs of leukemia even at 11 months. Perhaps not supported by the data as all the Y665F mice were euthanized at 6 months (24 weeks).

Response

Thank you for the opportunity to clarify. Experiments examining the possibility of leukemia development with age up until 11 months were performed in heterozygous *Stat5b*^{Y665F} and homozygous *Stat5b*^{Y665H} mice (Methods section, lines 525-528). The scientific premise for studying the heterozygous *Stat5b*^{Y665F} mice with age was the fact that the *STAT5B*^{Y665F} mutation in people occurs in the heterozygous state (Discussion section, line 425). We agree that study of homozygous *Stat5b*^{Y665F} mice would be a valuable contributing experiment to address its leukemic potential in mice, but we were unable to develop cohorts of aging homozygous *Stat5b*^{Y665F} mice due to the loss of mice due to dermatitis and, as discussed above, do not have the finances to address this in a transplant setting. We have clarified the limiting parameters of the presented experiments regarding leukemia development in the Results section (lines 242-245) and in the Discussion section (lines 425-430, 438-446).

Reviewer #3 (Comments to the Authors (Required)):

Lee et al. demonstrate in this paper that two different STAT5B mutations show opposing function by in vivo and in vitro experiment. Authors showed STAT5B-Y665F is a gain of function mutation in contrast to STAT5B-Y665H which shows loss of function. Authors examined the immune-phenotype of STAT5B mutation mouse and provided data to support the hypothesis that STAT5B-Y665F mutation could increase the STAT5B phosphorylation, transcriptional activities. The manuscript is very interesting, but several important concerns need to be addressed.

Major concerns:

1- Authors discovered two STAT5B mutations from human T cells leukemia and analyzed the predicated structural in silico. To test these two mutations' function, two mutation mouse models were introduced. Authors did not clarify if STAT5B-Y665 amino acid is conserved or not across different species. And which kind of STAT5B was transfected into mouse CD4T cells was not clear.

Response

Thank you for these points. We added text (Results section, lines 114-115) and a new figure (new Supplementary Figure 1) to indicate that the STAT5B-Y665 amino acid is 100% conserved across the range of 26 vertebrate species examined, including mammals, marsupials, birds, reptiles and amphibians. The kind of STAT5B transfected into the mouse CD4T cells has been clarified in the Methods section (lines 602-606).

2- In Fig2, authors transduced different mutants of STAT5B retrovirus vector into WT or Stat5a/b-DKO mice to test the function of STAT5B. According to the data showed in Fig2C, WT-STAT5B could not fully rescue the Stat5a/b DKO phenotype as showed by the CD25 expression level. Please explain the reason.

Response

We agree, retroviral reconstitution of 'normal' STAT5B (i.e. Y665) does not fully rescue CD25 expression in STAT5-deficient T cells (i.e. does not achieve WT CD25 and CD98 levels). We have clarified in the Methods section that the T cells are activated with anti-TCR and anti-CD28 antibodies prior to retroviral transduction (lines 602-603). Thus, WT cells are both producing and sensing IL-2 via STAT5 prior to transduction, while the KO cells can produce IL-2 but cannot sense it until STAT5B is restored. Put simply, the WT cells experience STAT5 signaling for a longer time span than the KOs.

3- In Fig3, authors showed increase activated CD8T cells and increase percentage of Treg cells in STAT5B-Y655F mice. What's the mechanism for increased both effector and regulatory cell subsets? Is this effect due to a cell-intrinsic or cell-extrinsic manner?

Response

Thank you for your question. Interestingly, STAT5 is known to play a role in the development, survival and function of both CD4⁺ regulatory T cells (see for example Fan et al. 2018) as well as CD8⁺ T effector cells (see for example Beltra et al. 2023). The presence of an activated STAT5B such as the STAT5B^{Y655F} might be hypothesized to impact populations of both activated CD8⁺ T cells as well as increasing CD4⁺ regulatory T cells, as observed here. If an activated STAT5B allele would be the mechanism for both types of cells to increase, then the effect would be deemed to be a cell-intrinsic effect. Pham et al. 2018 showed that the activated STAT5B mutant N642H was also associated with an increase in CD8⁺ T cell populations and that the effect was a cell-intrinsic mechanism as shown using bone marrow transplant models. Interestingly, their STAT5B mutant N642H also demonstrated significantly increased numbers of CD4⁺ T cells as we observed here. Because we were unable to perform bone marrow transplant studies, we cannot absolutely answer your question of whether or not there are cell extrinsic in addition to cell intrinsic effects. We have clarified this limitation in the Discussion section of the manuscript (lines 446-448).

4- IL2R(CD25) is the most important downstream gene regulated by STAT5B. It would be better to show the basic CD25 expression level in different mutant's mice model. And it is well known that naïve T cells are majorly CD25-negative. So it is meaningless to test pSTAT5 level after 30min *in vitro* stimulation, as CD25 is not expressed in naïve T cells.

Response

Thank you for raising this point. We have added the basal CD25 expression levels in T cells from the different mutant mouse models to Figure 5 (new panel C). Because three different interleukins (IL-2, IL-7, IL-15) were used to check pSTAT5 levels we present basal expression levels of the receptor components for all three interleukins. Different proportions of naïve CD4⁺ and CD8⁺ cells were present in the mice (Figures 3-4), (approximately 60-75% in the *Stat5b*^{Y665} and *Stat5b*^{Y665H} mice, and ~25% in the *Stat5b*^{Y665F} mice). This means that the population of cells tested with IL-2, IL-7 and IL-15 from all three mouse models included both naïve and non-naïve T cells. Consistent with the different activities of the three forms of STAT5b studied and the different proportions of naïve and non-naïve T cells in the different models, basal expression levels of the receptor components varied between models and between genes. As the reviewer suggests, expression levels of *Il2ra* (CD25) were significantly lower in the *Stat5b*^{Y665} and *Stat5b*^{Y665H} T cells as compared to *Stat5b*^{Y665F} T cells, and this could contribute to why only the *Stat5b*^{Y665F} T cells showed significant increases in pSTAT5 with a 30-minute IL-2 exposure. Interestingly *Il7r* was expressed at higher levels than *Il2ra* in all three models. With 30 minutes of IL-7 exposure both *Stat5b*^{Y665} and *Stat5b*^{Y665F} showed significant increases in pSTAT5. Only the *Stat5b*^{Y665H} derived T cells were unresponsive to all three interleukins. (It can be noted that this model exhibited very similar proportions of naïve and non-naïve T cells as compared to *Stat5b*^{Y665} mice.) In short, the reason for utilizing all three interleukins for a 30-minute *in vitro* stimulation was to illustrate the different levels of pSTAT5 without and with stimulation in the three models, recognizing the inherent limitations different basal expression levels of the receptors would pose. Importantly, we revised the Results section text (lines 275-291) to present the relative contributions of interleukin receptor levels and proportions of naïve T cells present to the STAT5b phosphorylation studies.

5- Authors used total T cells for histone modification and TF binding analysis. Considering the altered cell subsets, the difference showed in Fig 6, the majoring reason of the difference could be explained by different contributing from different cells subset but not its binding activities.

Response

Thank you. We agree that the results shown are reflective of both differences in ratios of different subsets as well as binding activities. We have clarified this point in the Results

section (line 296-307, 320-321). The discussion section (lines 422-424) has been edited to indicate a value of the comparative ChIP-seq study was identification of high-confident STAT5B binding peaks in the *Stat5b*^{Y665F} mice.

Minor concerns:

1- It is not convincing to use Q-PCR result indicate the expression level of retrovirus expression level. Western blot would be better to indicate similar level of proteins were introduced in Stat5a/b-DKO cells

Response

The data shown in Figure 2B are not qPCR but, rather, 'tags per million' (TPM) values sourced from the RNA-seq data presented in Figure 2D-G. We have clarified this in the Results section (line 172), Figure 2B, and the legend for Figure 2B.

2- It would be better to show the gating strategy in the Figures.

Response

The gating strategy for the FACs presented in Figure 3 is shown in new Supplementary Figure 4. Representative flow cytometry contour plots are shown in Figure 2. Representative histograms of the flow cytometry analysis (FACS) have been added to the revised Figure 5. Representative flow cytometry plots are shown in new Figure 7.

3- There are some typos, e.g. In Fig6D, "F665H" should be "Y655H". Please double check the manuscript.

Response

Thank you. We have corrected the typos in Figure 6D and double checked and corrected any others we found in the manuscript.

March 24, 2025

RE: Life Science Alliance Manuscript #LSA-2025-03222-TR

Dr. Lothar Hennighausen
National Institute of Diabetes and Digestive and Kidney Diseases
BG 8 RM 101
8 CENTER DR
Bethesda, MD 20892

Dear Dr. Hennighausen,

Thank you for submitting your revised manuscript entitled "STAT5B leukemic mutations, altering SH2 tyrosine 665, have opposing impacts on immune gene programs". We would be happy to publish your paper in Life Science Alliance pending final revisions necessary to meet our formatting guidelines.

- please be sure that the authorship listing and order is correct
- please upload all figure files as individual ones, including the supplementary figure files; all figure legends should only appear in the main manuscript file
- please add a Category for your manuscript in our system
- please add the X and Bluesky handles of your host institute/organization as well as your own or/and one of the authors in our system
- please be sure that the authorship listing and order are correct and match between the system and manuscript file
- please rename Summary to Abstract
- please consult our manuscript preparation guidelines <https://www.life-science-alliance.org/manuscript-prep> and make sure your manuscript sections are in the correct order
- please add Author Contributions for all authors to our system as well
- please add your main, supplementary figure, and table legends to the main manuscript text after the references section;
- we encourage you to revise the figure legends for Figure S5 such that the figure panels are introduced in alphabetical order
- please upload your Tables in editable .doc or excel format; They can be included at the bottom of the main manuscript file or be sent as separate files.
- please add callouts for Figures S2A-D; S5A-G and S6A-J to your main manuscript text

A. FINAL FILES:

-- Summary blurb (enter in submission system): A short text summarizing in a single sentence the study (max. 200 characters including spaces). This text is used in conjunction with the titles of papers, hence should be informative and complementary to the title. It should describe the context and significance of the findings for a general readership; it should be written in the

present tense and refer to the work in the third person. Author names should not be mentioned.

B. MANUSCRIPT ORGANIZATION AND FORMATTING:

Sincerely,

March 31, 2025

RE: Life Science Alliance Manuscript #LSA-2025-03222-TRR

Dr. Lothar Hennighausen
National Institute of Diabetes and Digestive and Kidney Diseases
BG 8 RM 101
8 CENTER DR
Bethesda, MD 20892

Dear Dr. Hennighausen,

Thank you for submitting your Research Article entitled "STAT5B leukemic mutations, altering SH2 tyrosine 665, have opposing impacts on immune gene programs". It is a pleasure to let you know that your manuscript is now accepted for publication in Life Science Alliance. Congratulations on this interesting work.

DISTRIBUTION OF MATERIALS:

Again, congratulations on a very nice paper. I hope you found the review process to be constructive and are pleased with how the manuscript was handled editorially. We look forward to future exciting submissions from your lab.

Sincerely,
